# Diversity-aware Weight Perturbation Promotes Robust Adaptation

**Zibo Chen** [1]   **Ruxin Li** [1]   **Zilu Wang** [1]

## Abstract

Compute-In-Memory (CIM) accelerators are promising for energy-efficient edge inference, yet they face fundamental challenges when deploying Deep Neural Networks (DNNs), as hardware-induced weight perturbations from intrinsic noise and device drift degrade accuracy and impede reliable inference. To tackle this challenge, we propose Diversity-aware Weight Perturbation (DWP), an immune-system-inspired training method that emulates affinity-based selection by exploiting sample-level prediction disagreement under diverse noise realizations to guide adaptive sample weighting, building robustness to weight perturbation. Experiments show that DWP-trained models consistently yield superior robustness, achieving over 15% accuracy improvements compared to standard-trained models under severe weight perturbations (mismatch level up to 70%) and maintaining inference accuracy at 90% over a simulated one-year CIM operation with only 2%-4% variation in accuracy. Moreover, under matched model and inference configurations, deployment on low-precision CIM hardware reduces inference energy by 38% compared to a GPU baseline. These results demonstrate that DWP enables robust and energy-efficient neural network deployment on resource-constrained edge devices with inherent hardware uncertainties.

## 1. Introduction

The rapid advancement of machine learning (Dosovitskiy, 2020) has enabled deep neural networks (DNNs) to achieve substantial performance gains (Hoffmann et al., 2022). However, modern DNNs typically entail massive parameter counts and intensive computation (Ambrogio et al., 2023),

---

[1]School of Intelligence Science and Engineering, Harbin Institute of Technology (Shenzhen), Shenzhen, 518055, China. Correspondence to: Zilu Wang <wangzilu@hit.edu.cn>.

*Proceedings of the 43rd International Conference on Machine Learning*, Seoul, South Korea. PMLR 306, 2026. Copyright 2026 by the author(s).

leading to significant energy consumption and latency overhead when deployed on conventional CPU/GPU/TPU architectures. This limitation conflicts with the stringent low-power and low-latency requirements (Wan et al., 2022) of edge devices in applications such as smart cameras, wearable devices, and drones. To address these challenges, researchers have explored alternative computing paradigms, including heterogeneous and neuromorphic architectures (Tsai et al., 2023; Yao et al., 2024; Mondal et al., 2024). In particular, memristors are non-volatile memory elements with programmable resistance states and constitute a core technology for Compute-In-Memory (CIM) hardware (Manning et al., 2025). By enabling in-situ data storage and key operations such as vector-matrix multiplication, memristor-based CIM can mitigate data-movement costs and offer a promising substrate for energy-efficient edge inference (Yu et al., 2025; Wang et al., 2025).

Deploying DNNs on edge hardware inevitably introduces heterogeneous noise, necessitating enhanced robustness to maintain operational reliability (Ma et al., 2024). Conventional mitigation strategies mainly rely on device calibration or redesigned training pipelines, which often incur high computational and engineering costs. (Ye et al., 2023). While empirical noise augmentation has been explored to improve robustness (Gokhale et al., 2022), such methods often exhibit restricted generalization (Zhao et al., 2022) and may not defend against out-of-distribution perturbations (Yao et al., 2022). In contrast, biological defense mechanisms, particularly immune systems, provide strong robustness and adaptivity under recurring perturbations (Chavlis & Poirazi, 2025). This motivates us to ask whether an immune-system-inspired learning principle can help DNNs adapt to recurring hardware-induced perturbations.

Following this line of inquiry, we draw upon principles from immunology to enhance DNN robustness. We propose **Diversity-aware Weight Perturbation (DWP)**, a robustness-oriented training framework that interprets hardware-induced noise as pathogenic antigens. DWP turns a single noise source into a diverse repertoire of weight perturbations (Gao et al., 2022), akin to how a single pathogen drives cellular lesions in diverse, stochastically bounded pathological directions (Stein et al., 2022). Specifically, we hypothesize that DNNs develop heightened robustness to dominant, prevalent perturbation features induced by noise,

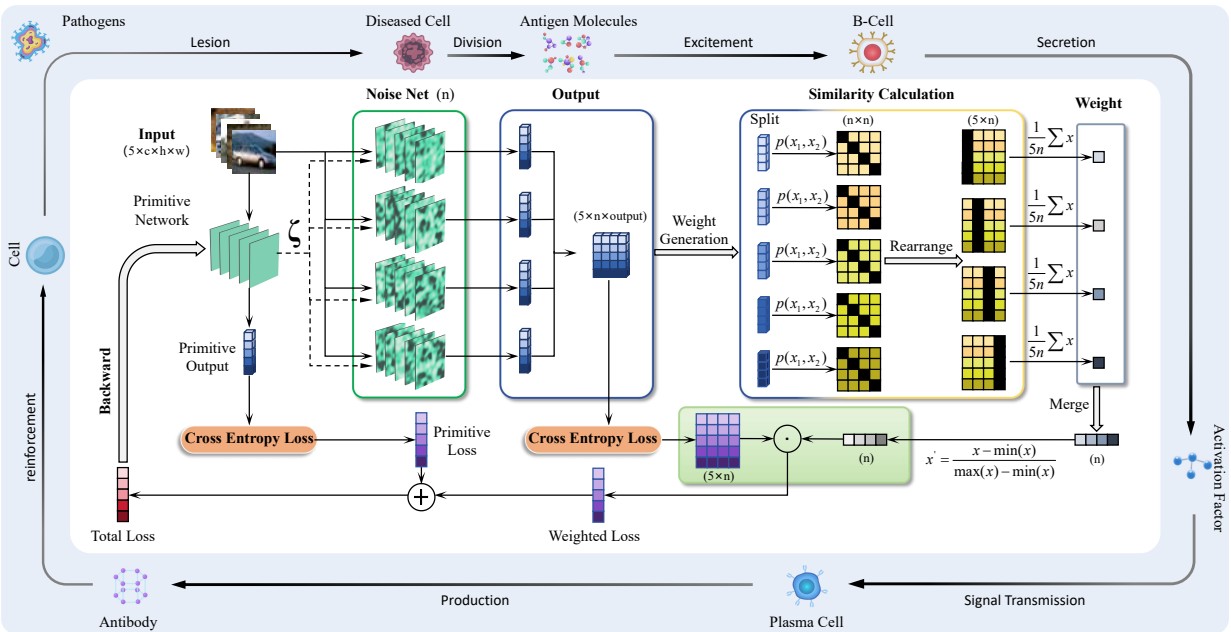

*Figure 1.* **Overview of DWP**. The outer blue cycle depicts the biomimetic analogy from pathogenic exposure to antibody reinforcement. Inside, an **Input** batch (batch size illustrated as 5 for clarity) is processed under an ensemble of noise-perturbed weight instances produced by **Noise Net** (shown with $n = 4$ for illustration). **Similarity Calculation** aggregates pairwise agreement (equivalently, low disagreement) among perturbed outputs to obtain an affinity-like score, and normalization yields adaptive **Weights** that modulate the robustness loss, yielding the total training objective.

prioritizing these features during training. This allows adaptation to different noise profiles, similar to adaptive immunity, resulting in models that are resilient to various perturbations and especially robust to recurring noise. In contrast, most existing techniques rely on regularized training to deal with noise in the worst case (Mao et al., 2023), which can be overly aggressive and cause the model to underemphasize the primary perturbation direction, deviating from the actual objective of robust training. We believe that immune-inspired defenses (Ren et al., 2024) offer a promising path toward long-term robustness under recurring hardware uncertainty. The contributions of this work are as follows:

- We propose DWP, an immune-system-inspired training method that emulates affinity-based selection by exploiting sample-level prediction disagreement under diverse noise realizations to guide adaptive weighting against weight perturbations.
- DWP achieves an absolute accuracy gain of over **15%** under severe weight perturbations (mismatch level=70%), surpassing prior robustness-enhancement methods in terms of robust generalization.
- We validate DWP in hardware-oriented edge settings. When deployed on a CIM simulator and a low-precision CIM hardware platform, DWP-trained models maintain **90%** accuracy over an equivalent continuous operation period of at least one year, and reduce inference energy by **38%** relative to a matched GPU baseline.

**Conflict of Interest Disclosure.** The authors declare no financial conflicts of interest related to this work.

## 2. Related Work

We review prior work on robustness under edge deployment constraints, focusing on parameter perturbations that are common in low-precision and CIM settings. Related methods mainly fall into three lines: weight-space adversarial training, noise-injection-based training, and adversarial regularization.

With the widespread adoption of smart terminals and IoT devices (Chang et al., 2021), the demand for edge AI deployments has surged and deep learning models are increasingly deployed on mobile and embedded platforms (Chen et al., 2020). However, limited compute and memory budgets often require aggressive compression and acceleration (Cordova-Cardenas et al., 2025), which can reduce robustness to both input and parameter perturbations (Awal et al., 2025). While adversarial training is widely used to improve robustness against input-space attacks (Ziras et al., 2025), recent studies show that perturbing network weights can cause incorrect predictions (Yu et al., 2023) and may be harder to diagnose in deployment (Dong et al., 2023; Özdenizci & Legenstein, 2022).

**Weight-space adversarial training:** Adversarial Weight

Perturbation (AWP) (Wu et al., 2020), applies worst-case perturbations to the weights during training. The motivation is that input-space adversarial training may still leave a sharp loss surface in the weight space, harming robust generalization. AWP searches a loss-maximizing weight perturbation and updates model parameters under the perturbed weights, forming a dual-perturbation training procedure. Adversarial Model Perturbation (AMP) (Zheng et al., 2021) similarly minimizes the worst-case empirical risk after perturbing all parameters within a bounded neighborhood, improving robustness to both parameter and input perturbations across architectures.

**Noise injection:** Noise injection during training provides an implicit regularizer that encourages parameter-insensitive representations. Injecting weighted noise during the forward pass (Murray & Edwards, 1994) improves robustness to parameter fluctuations and is particularly relevant to hardware-induced variations. We refer to this approach as Forward Noise.

**Adversarial regularization:** Adversarial Regularization (Büchel et al., 2022) introduces an explicit penalty on sensitivity to weight perturbations and performs iterative adversarial attacks in the weight space to approximate worst-case scenarios. This strategy improves robustness to target parameter changes and can enhance adaptability to hardware non-idealities. We refer to this approach as AWA.

Overall, existing methods either optimize for worst-case weight perturbations or enforce parameter insensitivity through noise/regularization. Yet they do not explicitly leverage cross-perturbation consensus to focus learning on dominant recurring interference patterns in edge deployment. We therefore propose DWP, an immune-system-inspired training method that improves robustness to recurring hardware-induced parameter perturbations in edge deployment.

## 3. Biomimetic Modeling

This section presents DWP, a biomimetic framework inspired by adaptive immune responses under repeated pathogenic exposure. As illustrated in Figure 1 and summarized in Algorithm 1, DWP follows an "exposure-affinity-reinforcement" logic. It first generates multiple stochastic weight-perturbed instances through **Noise Net** to emulate CIM weight-update mismatch. It then exploits sample-level prediction disagreement across diverse noise realizations in **Similarity Calculation** to obtain an affinity-like score for each perturbation instance. Finally, it converts these scores into normalized adaptive **Weights** that modulate the robustness loss during training. Below we detail the mathematical formulation step by step.

### Step 1: Pathogen Infiltration and Weight Infection

In DWP, hardware-induced perturbations are treated as

---

**Algorithm 1** Diversity-aware Weight Perturbation

> **begin**
>   $\Theta_i^* \leftarrow \Theta \odot (\mathbf{1} + \zeta \odot \mathbf{R}_i), \quad i \in \{1, \ldots, n\}$
>   **for** $i = 1$ **to** $n$ **do**
>     $\mathbf{H}_i \leftarrow f(\mathcal{X}; \Theta_i^*), \quad \mathcal{L}_i \leftarrow \text{Criterion}(\mathbf{H}_i, \mathbf{Y})$
>   **end**
>   **for** $i = 1$ **to** $n$ **do**
>     $\mathcal{S}_i \leftarrow \frac{1}{B(n-1)} \sum_{b=1}^{B} \sum_{j \neq i}^{n} \frac{\mathbf{h}_{b,i}^\top \mathbf{h}_{b,j}}{\|\mathbf{h}_{b,i}\|_2 \|\mathbf{h}_{b,j}\|_2}$
>   **end**
>   **for** $i = 1$ **to** $n$ **do**
>     $\omega_i \leftarrow \frac{\mathcal{S}_i - \min(\mathcal{S})}{\max(\mathcal{S}) - \min(\mathcal{S}) + \delta}$
>   **end**
>   $\Theta \leftarrow \Theta - \gamma \nabla_\Theta [\mathcal{L}_{\text{nat}}(\Theta, \mathcal{X}, \mathbf{Y}) + \sum_{i=1}^{n} \omega_i \mathcal{L}_i]$
> **end**

**Algorithm 1:** $\mathbf{R}_i \sim \mathcal{N}(0, 1)$; $\Theta, \Theta_i^*$: weights and perturbed weights (noise realization $i$); $B = |\mathcal{X}|$: batch size; $n$: number of noise realizations per iteration; $\mathcal{L}_i$: loss under $\Theta_i^*$; $\mathcal{L}_{\text{nat}}$: natural task loss under $\Theta$; $\mathcal{S}_i$: affinity-like agreement score aggregated from sample-level predictions across noise realizations; $\omega_i$: adaptive weight; $\gamma$: learning rate; $\zeta, \delta$: mismatch level and smoothing parameters.

---

pathogenic infiltration, and **Noise Net** generates diverse perturbed weight instances to emulate repeated exposure and stochastic mutation in immune responses. To model the random pathological effects induced by hardware uncertainty, we apply multiplicative Gaussian perturbations to the model parameters. We define the infected weights $\Theta_i^*$ resulting from the $i$-th pathogenic path as:

$$\Theta_i^* \leftarrow \Theta \odot (\mathbf{1} + \zeta \odot \mathbf{R}_i), \quad i \in \{1, \ldots, n\} \quad (1)$$

where $\zeta$ represents the level of mismatch that controls the magnitude of the perturbation, $\mathbf{R}_i \sim \mathcal{N}(0, 1)$ is a random variable sampled from a standard normal distribution and $\mathbf{n}$ represents the number of perturbed weight instances generated by **Noise Net** (i.e., $\mathbf{n}$ noise realizations sampled per iteration). In our main experiments, we set $\mathbf{n} = 10$ to balance robustness and training cost. Additional details are provided in Appendix D.

### Step 2: Antigenic Response and Batch Representation

Given an input batch, each infected weight instance induces a perturbed prediction profile, which we interpret as an antigenic response under a specific perturbation exposure. Specifically, for a batch of inputs $\mathbf{X} = \{\mathbf{x}_b\}_{b=1}^B$ with corresponding labels $\mathbf{Y}$, we evaluate the model under the $i$-th perturbed weight instance generated by **Noise Net** and compute the output batch $\mathbf{H}_i$ as:

$$\mathbf{H}_i = \{\mathbf{h}_{b,i}\}_{b=1}^B, \quad \text{where } \mathbf{h}_{b,i} = f(\mathbf{x}_b; \Theta_i^*) \in \mathbb{R}^K \quad (2)$$

Here, $\mathbf{h}_{b,i}$ denotes the logit vector for sample $b$ under the $i$-th perturbation. We compute the per-instance loss as $\mathcal{L}_i = \text{CE}(\mathbf{H}_i, \mathbf{Y})$. The resulting logits $\mathbf{h}_{b,i}$ form the basis

for measuring sample-level prediction disagreement across noise realizations in Step 3.

**Step 3: Global Immune Strength and Affinity Maturation**

Affinity maturation refers to the evolutionary process by which the immune system, through somatic hypermutation and clonal selection, enables antibodies to bind pathogens with higher strength. We transform this mechanism into a consensus-based feature selection logic: when a perturbation instance exhibits high cross-affinity with other instances, it indicates that it captures a dominant interference pattern shared across multiple noise realizations. **Similarity Calculation** operationalizes affinity-based selection by exploiting sample-level prediction disagreement across diverse noise realizations. Concretely, it computes pairwise cosine similarities (agreement) between perturbed logits and aggregates them over the batch. Higher cosine similarity indicates greater prediction agreement, indicating a more consistent interference pattern across realizations. The resulting global immune strength (affinity) score for the $i-th$ perturbation instance is defined as:

$$\mathcal{S}_i = \frac{1}{B(n-1)} \sum_{b=1}^{B} \sum_{\substack{j=1 \\ j \neq i}}^{n} \frac{\mathbf{h}_{b,i}^{\top} \mathbf{h}_{b,j}}{\|\mathbf{h}_{b,i}\|_2 \|\mathbf{h}_{b,j}\|_2} \tag{3}$$

By excluding self-correlation ($j \neq i$), $\mathcal{S}_i$ summarizes cross-realization agreement in the logit space. High $\mathcal{S}_i$ indicates that the $i$-th perturbation instance yields predictions consistent with the ensemble, suggesting alignment with a dominant interference pattern shared across noise realizations.

**Step 4: Antibody Secretion and System Reinforcement**

To ensure the stability of model updates, affinity scores are converted into normalized adaptive **Weights** derived from sample-level predictions, much like the activation factor in biological mechanisms:

$$\omega_i = \frac{\mathcal{S}_i - \min(\boldsymbol{\mathcal{S}})}{\max(\boldsymbol{\mathcal{S}}) - \min(\boldsymbol{\mathcal{S}}) + \delta} \tag{4}$$

where $\delta$ is a smoothing parameter ensuring numerical stability. Finally, our algorithm optimizes a natural (task) loss and a separate robustness loss:

$$\mathcal{L}_{\text{total}}(\boldsymbol{\Theta}) = \mathcal{L}_{\text{nat}}(\boldsymbol{\Theta}, \mathcal{X}, \mathbf{Y}) + \sum_{i=1}^{n} \omega_i \mathcal{L}_i(\boldsymbol{\Theta}_i^*, \mathcal{X}, \mathbf{Y}) \tag{5}$$

This step reinforces robust learning by prioritizing perturbation instances with higher affinity-like consensus, analogous to stronger antibody secretion in immune responses.

In summary, **DWP** promotes robust adaptation by being explicitly diversity-aware. It repeatedly exposes the model

to diverse stochastic weight perturbations, exploits sample-level prediction disagreement across noise realizations to perform affinity-like selection, and reinforces robustness through adaptive weighting in the training objective. To provide a formal characterization of stability, we model the loss using a second-order Taylor expansion, thereby connecting the empirical curvature coefficient $k$ to the directional Hessian projection $(v^{\top} H v)$. This theoretical analysis suggests that DWP improves robustness by reducing sensitivity to unstable perturbation directions and steering the optimization toward a flatter local manifold. We emphasize that this analysis should be interpreted as a curvature-based explanation of the cosine-consensus weighting rule, rather than a strict theoretical proof that diversity-aware weighting explicitly diagonalizes the perturbation covariance. For a comprehensive analytical derivation and further discussion regarding this mapping, please refer to Appendix A.

## 4. Evaluation

This section presents a comprehensive experimental evaluation of DWP to validate its efficacy in enhancing model robustness. Our experimental roadmap is structured into four primary objectives: 1) a comparative analysis of DWP against other optimization algorithms; 2) ablation studies to verify the necessity of the bio-inspired components; 3) evaluate the generalization ability of DWP under different model architectures; and 4) a validation of its robustness in CIM simulators and physical hardware deployments.

### 4.1. Experimental Setup

**Dataset:** We evaluated our method on multiple tasks. For **computer vision**, we used F-MNIST (CNN), CIFAR10 (ResNet (He et al., 2016), BiLSTM (Graves & Schmidhuber, 2005), ViT (Dosovitskiy, 2020)), and LFW (CNN); for **bio-signals**, we used ECG (LSTM); and for **audio**, US8K (CNN). Further dataset and architecture details are in Appendix E and additional materials. To further address the concern on experimental scale, we additionally evaluate DWP on the iFLYTEK 119 with Qwen3, covering a larger-scale dataset and model setting. The detailed experimental setup and results are provided in Appendix G.

To ensure a rigorous and fair evaluation (Pineau et al., 2021), the models are trained with a specific mismatch level $\zeta_1$ and the trained model is tested with $\zeta_2$. Robustness is quantified by the mean test accuracy and standard deviation calculated across 50 independent trials. We define the **baseline** as a model trained in a **noise-free** environment without any robustness enhancement. To isolate the impact of the proposed algorithms, all experiments within the same dataset and architecture maintain identical base hyperparameters.

We do not assume that the training noise distribution is iden-

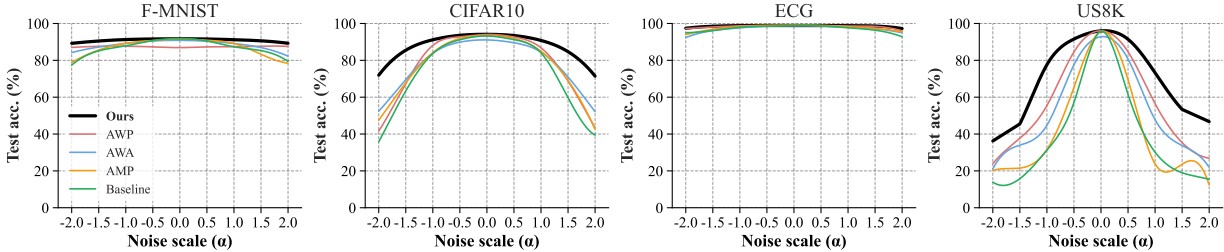

*Figure 2.* **Test accuracy landscape under cross-mismatch evaluation**. We use mismatch level $\zeta_1 = 0.2$ for training and $\zeta_2 = \alpha\zeta_1$ for testing. With the shift of $\alpha$, we find that the model trained using DWP is able to find **higher values in flatter locations**. It can be seen that the model trained by DWP has **better convergence** (more experimental results can be found in Figure A1 of the Appendix).

*Table 1.* Performance across tasks and architectures under different weight-update mismatch levels $\zeta$ (i.e., extent of hardware-induced perturbation). We compare DWP with baseline training and prior robustness methods (AWA, AWP, Forward Noise, AMP). Bold text indicates DWP experimental results, and underlined text indicates the best performing individual.

**F-MNIST** CNN — Mean Acc. / Std. (%)

| Mismatch ($\zeta$) | Ours* | Baseline | AWA | AWP | Forward Noise | AMP |
|---|---|---|---|---|---|---|
| Standard (0.0) | **91.56 / 0.00** | 91.79 / 0.00 | 91.83 / 0.00 | 87.46 / 0.00 | 91.70 / 0.00 | 91.62 / 0.00 |
| 0.1 | **91.20 / 0.13** | 90.53 / 0.54 | 91.02 / 0.23 | 86.79 / 0.44 | 91.74 / 0.31 | 91.05 / 0.31 |
| 0.2 | **91.23 / 0.17** | 87.98 / 1.75 | 88.97 / 0.92 | 87.57 / 0.44 | 91.63 / 0.19 | 88.67 / 1.41 |
| 0.3 | **90.74 / 0.61** | 82.19 / 4.09 | 87.31 / 1.34 | 87.85 / 0.47 | 90.81 / 0.20 | 85.24 / 2.69 |
| 0.4 | **90.71 / 0.30** | 77.31 / 4.87 | 84.52 / 2.30 | 86.65 / 0.79 | 90.06 / 0.40 | 82.37 / 2.98 |
| 0.5 | **90.45 / 0.38** | 66.45 / 8.62 | 82.17 / 2.94 | 85.47 / 1.83 | 88.04 / 0.88 | 71.18 / 6.74 |
| 0.6 | **90.36 / 0.39** | 61.73 / 8.31 | 76.33 / 6.58 | 80.48 / 6.20 | 85.76 / 1.38 | 62.58 / 9.46 |
| 0.7 | **90.01 / 0.44** | 53.95 / 9.87 | 71.40 / 7.91 | 73.94 / 10.24 | 81.49 / 2.49 | 60.50 / 8.10 |

**ECG** LSTM — Mean Acc. / Std. (%)

| Mismatch ($\zeta$) | Ours* | Baseline | AWA | AWP | Forward Noise | AMP |
|---|---|---|---|---|---|---|
| Standard (0.0) | **98.91 / 0.00** | 98.96 / 0.00 | 98.93 / 0.00 | 98.86 / 0.00 | 99.04 / 0.00 | 98.91 / 0.00 |
| 0.1 | **98.94 / 0.05** | 98.85 / 0.12 | 98.73 / 0.13 | 98.86 / 0.04 | 98.94 / 0.04 | 98.89 / 0.11 |
| 0.2 | **98.78 / 0.14** | 98.06 / 0.75 | 97.98 / 0.60 | 98.72 / 0.18 | 98.88 / 0.09 | 98.15 / 0.74 |
| 0.3 | **98.69 / 0.23** | 96.58 / 2.53 | 96.87 / 1.87 | 97.93 / 0.78 | 98.61 / 0.30 | 94.36 / 4.12 |
| 0.4 | **98.54 / 0.40** | 94.28 / 3.20 | 96.24 / 1.78 | 96.33 / 1.75 | 98.09 / 0.77 | 94.53 / 3.24 |
| 0.5 | **98.19 / 0.87** | 91.05 / 5.11 | 95.25 / 2.62 | 93.76 / 3.02 | 96.88 / 1.50 | 90.18 / 6.76 |
| 0.6 | **97.87 / 1.18** | 87.29 / 7.51 | 93.28 / 2.74 | 92.57 / 3.42 | 95.00 / 1.86 | 85.80 / 8.12 |
| 0.7 | **97.23 / 1.73** | 86.08 / 6.37 | 92.87 / 2.11 | 89.63 / 3.71 | 92.75 / 3.73 | 79.03 / 13.51 |

**US8K** CNN — Mean Acc. / Std. (%)

| Mismatch ($\zeta$) | Ours* | Baseline | AWA | AWP | Forward Noise | AMP |
|---|---|---|---|---|---|---|
| Standard (0.0) | **99.77 / 0.00** | 95.69 / 0.00 | 98.31 / 0.00 | 92.52 / 0.00 | 87.19 / 0.00 | 97.38 / 0.00 |
| 0.1 | **87.35 / 8.15** | 60.83 / 13.28 | 77.82 / 9.59 | 83.53 / 5.53 | 82.92 / 2.52 | 65.73 / 12.24 |
| 0.2 | **72.99 / 10.08** | 28.02 / 10.41 | 53.97 / 12.35 | 54.80 / 8.70 | 77.85 / 5.31 | 32.19 / 9.98 |
| 0.3 | **61.29 / 11.91** | 19.31 / 5.90 | 36.61 / 11.25 | 38.60 / 7.36 | 61.38 / 10.37 | 22.79 / 6.38 |
| 0.4 | **55.26 / 11.26** | 15.52 / 4.44 | 27.20 / 8.77 | 22.96 / 8.13 | 48.95 / 13.11 | 15.73 / 4.74 |
| 0.5 | **48.20 / 12.30** | 12.25 / 3.80 | 22.66 / 7.54 | 20.69 / 7.23 | 37.47 / 11.53 | 14.12 / 4.16 |
| 0.6 | **43.56 / 10.66** | 12.88 / 2.70 | 20.37 / 6.77 | 15.62 / 4.71 | 28.78 / 8.90 | 12.11 / 2.98 |
| 0.7 | **39.70 / 10.40** | 11.38 / 3.20 | 18.01 / 6.08 | 14.05 / 4.24 | 23.79 / 7.52 | 11.85 / 3.43 |

**CIFAR10** ResNet — Mean Acc. / Std. (%)

| Mismatch ($\zeta$) | Ours* | Baseline | AWA | AWP | Forward Noise | AMP |
|---|---|---|---|---|---|---|
| Standard (0.0) | **93.02 / 0.00** | 92.75 / 0.00 | 92.75 / 0.00 | 93.90 / 0.00 | 93.28 / 0.00 | 93.22 / 0.00 |
| 0.1 | **92.37 / 0.46** | 91.04 / 0.54 | 90.82 / 0.87 | 92.19 / 0.61 | 91.40 / 0.64 | 91.08 / 0.60 |
| 0.2 | **90.22 / 1.30** | 79.96 / 5.67 | 82.29 / 3.55 | 83.88 / 4.29 | 86.52 / 3.38 | 82.64 / 3.40 |
| 0.3 | **87.54 / 3.09** | 57.52 / 9.58 | 66.35 / 5.59 | 61.91 / 10.16 | 76.45 / 5.96 | 57.17 / 9.91 |
| 0.4 | **83.51 / 3.43** | 30.47 / 8.72 | 43.40 / 10.93 | 28.90 / 9.25 | 62.65 / 9.82 | 25.29 / 8.14 |
| 0.5 | **73.00 / 8.70** | 18.02 / 4.60 | 28.52 / 9.53 | 19.06 / 5.11 | 45.01 / 9.65 | 17.20 / 4.33 |
| 0.6 | **66.48 / 9.61** | 12.36 / 2.40 | 15.63 / 4.43 | 14.10 / 2.97 | 30.84 / 10.16 | 12.32 / 2.36 |
| 0.7 | **65.61 / 10.59** | 10.65 / 1.21 | 12.06 / 2.60 | 10.78 / 1.73 | 22.57 / 7.04 | 11.32 / 1.71 |

**CIFAR10** BiLSTM — Mean Acc. / Std. (%)

| Mismatch ($\zeta$) | Ours* | Baseline | AWA | AWP | Forward Noise | AMP |
|---|---|---|---|---|---|---|
| Standard (0.0) | **75.49 / 0.00** | 71.20 / 0.00 | 74.54 / 0.00 | 73.96 / 0.00 | 69.98 / 0.00 | 74.40 / 0.00 |
| 0.1 | **73.96 / 0.26** | 69.41 / 1.28 | 72.63 / 0.81 | 72.34 / 0.67 | 71.12 / 0.14 | 71.83 / 1.55 |
| 0.2 | **72.49 / 0.30** | 64.14 / 3.84 | 69.16 / 1.52 | 66.60 / 5.53 | 69.99 / 0.30 | 67.72 / 3.36 |
| 0.3 | **71.19 / 0.47** | 56.56 / 6.84 | 63.69 / 3.62 | 59.62 / 8.49 | 69.61 / 0.41 | 58.48 / 7.75 |
| 0.4 | **69.87 / 0.53** | 48.15 / 9.71 | 57.07 / 5.30 | 54.31 / 8.63 | 67.60 / 0.77 | 49.00 / 10.14 |
| 0.5 | **69.20 / 0.61** | 41.37 / 9.23 | 47.78 / 6.42 | 46.14 / 10.04 | 67.01 / 1.28 | 40.37 / 10.75 |
| 0.6 | **67.64 / 0.54** | 38.26 / 9.51 | 42.45 / 7.83 | 39.51 / 11.06 | 63.50 / 2.76 | 34.67 / 10.44 |
| 0.7 | **67.32 / 0.53** | 32.15 / 8.65 | 34.54 / 9.70 | 32.13 / 10.39 | 59.19 / 4.54 | 29.69 / 8.88 |

**CIFAR10** ViT — Mean Acc. / Std. (%)

| Mismatch ($\zeta$) | Ours* | Baseline | AWA | AWP | Forward Noise | AMP |
|---|---|---|---|---|---|---|
| Standard (0.0) | **76.85 / 0.00** | 77.12 / 0.00 | 76.35 / 0.00 | 76.35 / 0.00 | 77.25 / 0.00 | 77.04 / 0.00 |
| 0.1 | **76.21 / 0.21** | 75.79 / 0.21 | 76.34 / 0.20 | 75.95 / 0.21 | 76.64 / 0.15 | 75.84 / 0.24 |
| 0.2 | **75.46 / 0.33** | 73.19 / 0.41 | 73.98 / 0.29 | 73.88 / 0.34 | 76.01 / 0.28 | 73.66 / 0.35 |
| 0.3 | **74.75 / 0.33** | 67.43 / 0.74 | 69.84 / 0.42 | 69.76 / 0.70 | 74.12 / 0.28 | 68.91 / 0.93 |
| 0.4 | **73.06 / 0.46** | 57.24 / 1.85 | 63.82 / 0.62 | 61.20 / 1.34 | 71.16 / 0.51 | 60.63 / 1.37 |
| 0.5 | **71.10 / 0.48** | 43.90 / 2.65 | 58.75 / 0.78 | 47.95 / 2.31 | 66.06 / 1.06 | 47.12 / 1.91 |
| 0.6 | **68.70 / 0.55** | 29.45 / 2.81 | 51.70 / 1.44 | 34.96 / 2.94 | 56.21 / 1.83 | 33.21 / 2.89 |
| 0.7 | **65.30 / 0.72** | 20.15 / 2.08 | 45.24 / 2.03 | 22.36 / 2.67 | 44.04 / 3.04 | 22.67 / 2.44 |

tical to the actual CIM hardware noise distribution; instead, it serves as a controllable proxy for deployment-time weight deviations caused by device mismatch, programming errors, and temporal drift (Büchel et al., 2022). We therefore adopt a cross-mismatch protocol, where models trained with a perturbation level $\zeta_1$ are tested under different perturbation levels $\zeta_2$, to evaluate robustness transfer under mismatched deployment noise conditions.

## 4.2. Performance Benchmarking

For fair comparison, we follow the original settings of each method: AWA uses $\beta_{\text{rob}}$=0.25, $N_{\text{steps}}$=10; AWP adopts $\gamma$=0.01 with $(K2, A)$=$(1, 1)$; AMP sets $\epsilon$=0.005 with one normalized-gradient update ($N$=1); and Forward Noise applies dropout with probability 0.3, consistent with AWA's comparison setting (Büchel et al., 2022).

**Weight Landscape and Robustness:** While prior work has leveraged weight loss landscapes to evaluate robustness (Wu et al., 2020), we instead employ **weight-accuracy landscapes** for a more intuitive assessment of inference stability under perturbations. A high-curvature ("sharp")

landscape indicates strong sensitivity to parameter changes, where even minor perturbations can cause severe accuracy degradation. In contrast, a low-curvature ("flat") landscape suggests robustness, with the model maintaining stable performance despite weight drift.

**Flatness of the weight-accuracy landscape:** We analyzed the test weight-accuracy landscape on four datasets as shown in Figure 2. Following a rigorous evaluation protocol, we perturbed the trained weights $\Theta$ with a random vector $v \sim \mathcal{N}(0, \zeta|\Theta|)$, where $\zeta = 0.2$ and recorded the test accuracy over a wide range of noise scales $\alpha \in [-2, 2]$.

**Results and Analysis:** The results highlight the clear superiority of our method. Whereas the baseline and other optimization approaches exhibit sharp parabolic drops in accuracy with increasing mismatch levels, DWP maintains a consistently **flat** and **elevated** accuracy profile across all tasks. Under extreme perturbation ($\alpha = \pm 2$), DWP achieves substantial gains over the baseline: on **F-MNIST**, test accuracy improves from 77.0% to 89.0% (+12%); on **CIFAR10**, from 38.0% to 71.0%, surpassing the next-best method by at least 14%. On the **ECG** task, accuracy rises from 88.0% to 96.0%, matching noise-free conditions even under severe

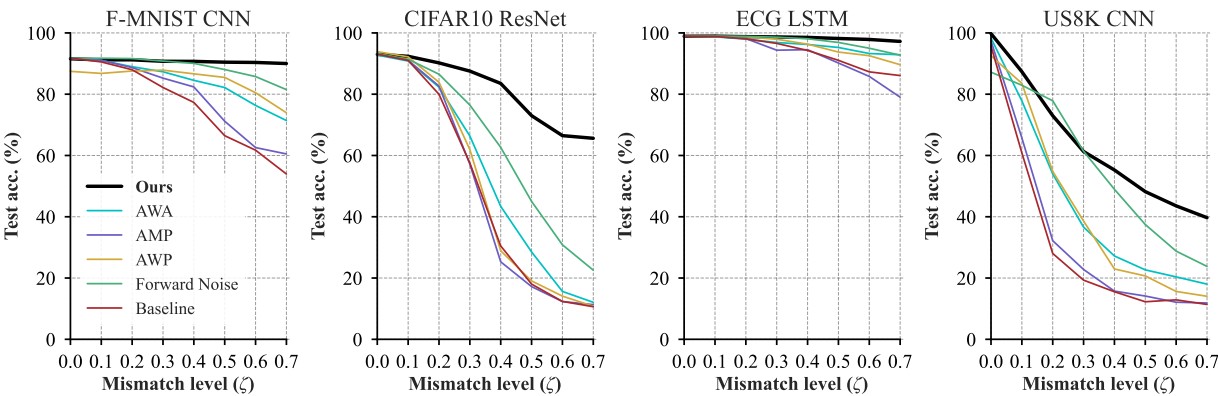

*Figure 3.* **Performance of the DWP-trained model under attacks of varying intensity.** The model trained using DWP showed the best performance compared to other trained models under different mismatch levels of attacks. In this experiment, the training mismatch level and the testing mismatch level are consistent: $\zeta_1 = \zeta_2$.

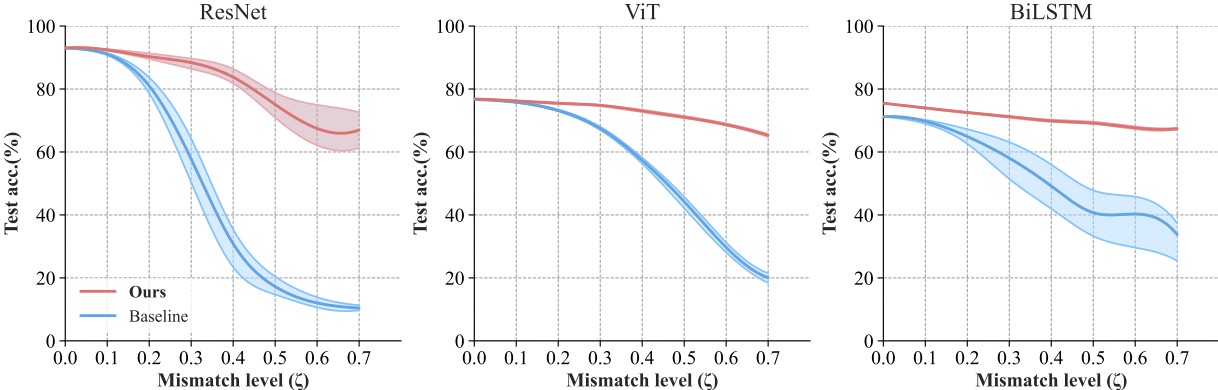

*Figure 4.* **Cross-architecture robustness under increasing mismatch.** It can be seen that the proposed DWP (red manifold) consistently outperforms the baseline (blue manifold) across ResNet, ViT, and BiLSTM architectures under increasing mismatch level ($\zeta$). Shaded regions delineate the **first and third quartiles** (25th–75th percentiles) of test accuracy distributions. This representation highlights data concentration and mitigates the influence of statistical outliers. See Table 1 for the specific data.

perturbations. For the challenging **US8K** dataset, DWP delivers a 21% boost ($16.0\% \rightarrow 37.0\%$), consistently outperforming alternatives.

**Robustness Against Adversarial Task Perturbations:** Figure 3 and Table 1 present a cross-dataset comparison of DWP with baseline and alternative optimization methods under direct perturbation of task performance. We define a 60% accuracy threshold as the boundary of **severe functional disruption**. While most competing methods attempt to regularize against worst-case perturbations, they exhibit poor generalization as mismatch intensity $\zeta$ increases. In contrast, DWP consistently achieves broad-spectrum robustness, as reflected by shallower accuracy decay and higher worst-case performance across all tasks.

For example, On **F-MNIST**, DWP sustains near-optimal accuracy ($90.01\%$) even at maximum mismatch ($\zeta=0.7$), outperforming the baseline ($53.95\%$) by $36.06\%$, with far lower variance (std 0.44 vs. 9.87). On **ECG**, it preserves near-perfect functionality across the entire noise spec-

trum ($98.91\% \rightarrow 97.23\%$). For the challenging **CIFAR10** (ResNet), DWP is the only method to avoid catastrophic failure at $\zeta=0.5$, maintaining $73.00\%$ accuracy, while AMP, AWP, and the Baseline collapse to $17.20\%$, $19.06\%$, and $18.02\%$, respectively. The robustness gap is even more pronounced on US8K: when $\zeta < 0.3$, DWP eliminates severe functional disruption, and its accuracy at high mismatch levels ($\zeta = 0.7$) exceeds that of other methods by at least 10%.

Actually, the strikingly flat weight-accuracy landscapes in Figure 2 and the performance gains in Figure 3 stem directly from the regularization term $\sum_{i=1}^{n} \omega_i \mathcal{L}_i(\mathbf{\Theta}_i^*, \mathcal{X}, \mathbf{Y})$. By assigning diversity-aware weights $\omega_i$, our framework prevents stochastic exploration from collapsing into narrow, low-dimensional subspaces. This promotes the suppression of sharp curvature directions in the loss surface, effectively steering the model toward the broad, stable "Basin Minimum" observed in empirical evaluations. While DWP incurs a modest overhead due to multiple forward propagations, this cost is justified by significant gains in robust

generalization. The method remains scalable, as detailed in Appendix C. Additional analysis on the training cost, convergence behavior, resource overhead, worst-case accuracy, 10th-percentile accuracy, and failure probability of DWP is provided in the Additional Material.

### 4.3. Architectural Versatility

**The Necessity of Cross-Architecture Validation:** To establish DWP as a fundamentally robust optimization paradigm, it is essential to evaluate its effectiveness across heterogeneous architectures (Premchandar et al., 2022). Different models exhibit distinct inductive biases, and a truly generalizable defense should be architecture-agnostic—its robustness must stem from principled regularization of the weight space. By validating DWP across diverse network types, we demonstrate that its anti-interference effect is universal: it suppresses perturbations directly at the weight level, independent of the underlying model structure. This validation is vital for edge deployment, ensuring robustness across diverse architectures and complex tasks.

To rigorously evaluate the **architectural agnosticism** and generalization capacity of DWP under **model heterogeneity**, we conducted comprehensive experiments on the **CIFAR10** dataset using three fundamentally distinct architectures: ResNet (convolutional), ViT (attention-based), and BiLSTM (recurrent). While we acknowledge that certain models, particularly BiLSTM applied to image data, do not achieve state-of-the-art accuracy due to inherent misalignment between model inductive biases and data modality, our goal here is not to benchmark peak performance but to isolate and examine the **intrinsic robustification effect** introduced by DWP.

Specifically, as shown in Figure 4, DWP (red manifold) consistently yields a flatter accuracy landscape than the **baseline** (blue manifold) across all architectures. Even under extreme perturbation ($\zeta = 0.7$), where the baseline suffers catastrophic degradation, DWP preserves substantial functional integrity. These results highlight DWP's universal ability to stabilize decision boundaries irrespective of architectural priors, reinforcing its model-agnostic robustness.

### 4.4. Ablation Analysis

We assess the individual contributions of DWP components via ablation experiments, as shown in Figure 5. Our first observation is that injecting multiplicative weight noise alone (denoted as the **Noise group**) already yields substantial robustness over the **Baseline**. For instance, on **CIFAR10** at $\zeta = 0.7$, accuracy improves from 12.0% to 50.0%, demonstrating that weight-space perturbation forms a necessary foundation for defending against hardware-induced noise.

However, as shown in the left subfigure of Figure 5, this naive approach degrades quickly as mismatch intensifies. The primary robustness gain stems from the proposed **diversity-aware selection**, which consistently enhances performance beyond the Noise group. On CIFAR10, **DWP** achieves an additional 14.0% accuracy boost, indicating that random noise exposure captures only partial robustness, while structured selection reinforces generalizable features through consensus alignment.

In addition to higher mean accuracy, DWP offers markedly improved statistical stability. As shown in the middle subfigure of Figure 5, the Noise group exhibits rising variance under stronger perturbations, reflecting sensitivity to specific noise realizations. In contrast, DWP maintains consistently low variance through affinity-based weighting, which suppresses atypical noise directions and guides optimization toward a more stable region of the loss surface. This effect is further illustrated in the right subfigure of Figure 5: while noise injection alone marginally flattens the weight-accuracy landscape, DWP induces a more uniformly stable manifold. By converting unstructured noise into structured generalization via diversity selection and affinity aggregation, DWP enables reliable performance even under extreme hardware mismatch. Additional sensitivity analysis on key hyperparameters, including the training perturbation level, noise realization number, normalization parameter, and learning rate, is provided in the Additional Material.

### 4.5. CIM Implementation

*Table 2.* Deployment comparison of the DWP-trained model on GPU and CIM platforms.

| Metric | GPU | CIM |
|---|---|---|
| Accuracy (%) | $88.50 \pm 0.00$ | $82.93 \pm 1.41$ |
| Average Latency | 0.9436 ms | 0.27 s |
| Real-time Compute | 1.1462 GFLOPS | 0.004245 GOPS |
| Power | 51.33 W | 0.135 W |
| Average Energy | 48.43 mJ | 29.97 mJ |
| Average Clock Speed | 2490 MHz | 30 MHz |

**Robustness Against Temporal Drift in Memristor-based CIM:** CIM architectures based on memristor devices present a compelling solution for energy-efficient edge inference. However, their practical deployment is challenged by **temporal parameter drift**—a stochastic process intrinsic to device physics that causes gradual degradation of stored weights over time (Freye et al., 2022).

To further evaluate DWP's effectiveness under such conditions, we simulated long-term inference on the **LFW** face recognition task using a specialized CIM simulator. As shown in Figure 6, we tracked the test accuracy and variance over a one-year period ($T_{\inf} \approx 3.15 \times 10^7$ s) for models trained with varying mismatch levels $\zeta \in \{0.4, 0.5, 0.6, 0.7\}$.

The experimental results in Figure 6 reveal that DWP-

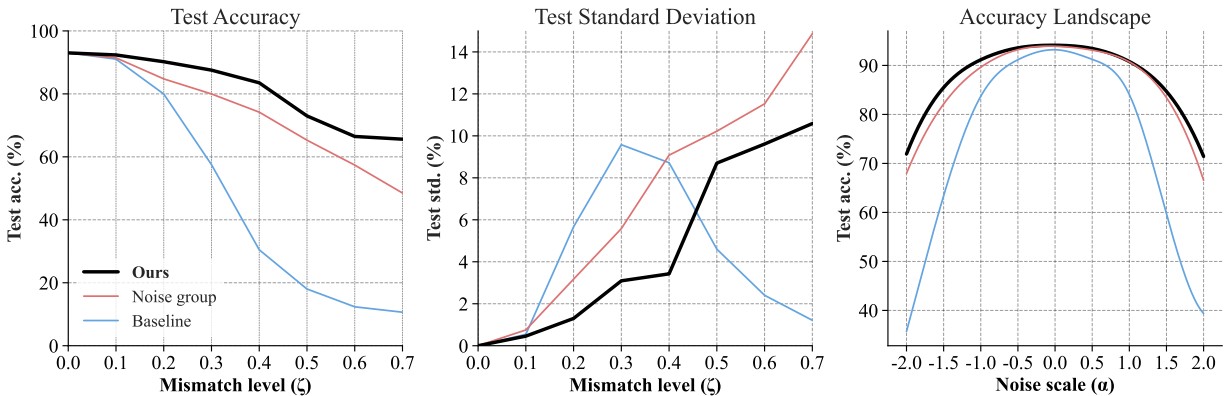

*Figure 5.* **Performance metrics for ResNet on CIFAR10 across mismatch levels.** Removing the biologically-motivated diversity-selection mechanism harms the overall functional robustness and stability of DWP.

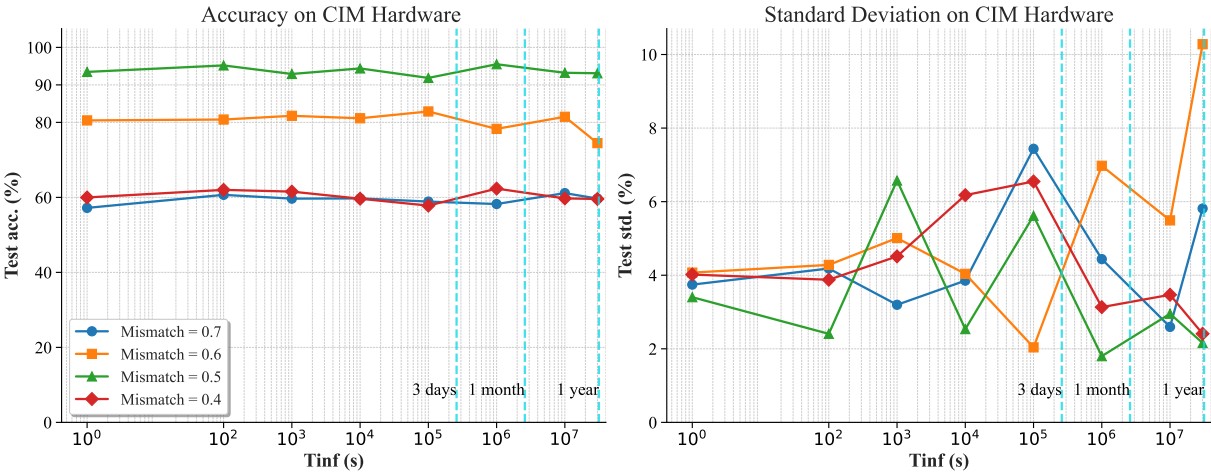

*Figure 6.* **Year-long robustness of DWP under CIM drift.** We deploy DWP-trained models (trained at different mismatch levels) to a CIM simulator and track recognition performance on LFW face dataset over an equivalent one-year CIM operation period.

trained models exhibit sustained robustness over time, with accuracy degradation limited to within 2-4% and variance remaining consistently low throughout the year. In contrast, baseline models, which lack targeted perturbation exposure, collapse immediately under drift-induced noise.

Crucially, the $\zeta = 0.5$ configuration (green curve) achieves the best long-term performance, maintaining $> 90\%$ accuracy and the lowest variance. This indicates that robustness is maximized when the training-time mismatch level closely matches the statistical properties of deployment-time drift. By proactively exposing the model to representative noise directions during training, DWP effectively aligns the learned weight manifold with the drift dynamics of CIM hardware.

These results demonstrate that DWP not only improves short-term robustness but also enables **long-term reliable deployment** of DNNs on noisy hardware—an essential property for sustainable edge intelligence.

**Deployment on Low-Precision CIM Hardware:** To assess the real-world viability of our framework under resource-constrained conditions, we deployed the DWP-trained

model on a physical CIM substrate for **FashionMNIST** classification, with all weights quantized to **4-bit precision**.

The experimental results in Table 2 shown that the CIM-deployed model achieves an accuracy of $82.93\% \pm 1.41\%$, only $5.57\%$ lower than the 32-bit GPU baseline ($88.50\%$), despite aggressive quantization and intrinsic device-level non-idealities such as conductance variation. This result highlights the robustness of DWP-trained models under extreme hardware constraints.

Crucially, CIM offers substantial energy advantages. Without any low-level algorithmic tuning or hardware-specific optimization, the CIM system operates at just $0.135$ W, which is over $380\times$ lower than the GPU's $51.33$ W, and it reduces average inference energy from $48.43$ mJ to $29.97$ mJ. While CIM's compute throughput and clock speed are lower ($0.004245$ GOPS @ $30$ MHz), the energy efficiency gain significantly outweighs this trade-off in edge scenarios.

To further demonstrate the advantage of DWP for robust low-precision CIM deployment, we also deploy models trained by other robustness-enhancement methods on the

same physical CIM hardware platform. All models are evaluated under the same 4-bit mapping and identical device conditions, so that the comparison focuses on algorithmic robustness rather than differences in hardware configuration. As shown in Table 3, DWP achieves the highest inference accuracy on the low-precision CIM platform, outperforming Forward Noise, AWA, AWP, and AMP. These results indicate that DWP provides stronger robustness under practical low-precision and hardware-noisy deployment conditions.

These results demonstrate that combining DWP with CIM hardware offers a promising path toward sustainable, robust, and energy-efficient edge intelligence. We further conduct matched deployment comparisons between the CIM simulator and the physical hardware platform, showing that the comparable inference accuracy between the two platforms provides empirical support for the reasonableness of the simulator. Details of the CIM simulator, the low-precision CIM platform, and the supplementary alignment experiments are provided in the additional materials and Appendix B.

*Table 3.* Comparison on the same physical low-precision CIM platform.

| Method | CIM Accuracy (%) |
|---|---|
| DWP (Ours) | **82.93 ± 1.41** |
| Forward Noise | 71.45 ± 6.30 |
| AWA | 64.90 ± 6.40 |
| AWP | 64.50 ± 7.20 |
| AMP | 56.90 ± 1.50 |

## 5. Conclusion

We propose a biomimetic training paradigm called DWP, which draws inspiration from immune affinity selection and utilizes diversity-aware stochastic weight perturbations with adaptive weighting to improve robustness. Theoretically, our analysis based on the Hessian matrix shows that DWP can systematically suppress high curvature regions, thereby reducing the model's sensitivity to parameter drift. Extensive experiments on heterogeneous architectures and multimodal datasets (including images, ECGs, and audio) demonstrate that models trained using DWP consistently maintain higher accuracy and predictive stability as the level of perturbation increases, highlighting its architectural independence and ensuring its broad applicability across different network types. Crucially, we validate the practical feasibility of DWP via deployment on physical low-precision CIM hardware. Under 4-bit quantization, the DWP-trained model maintains high accuracy while reducing inference energy by nearly half compared to traditional GPUs. Although DWP increases computational overhead during the training phase, it delivers substantial robustness and deployment-level efficiency gains, making it a practical enabler for robust and energy-efficient edge inference.

## Acknowledgements

We thank our anonymous reviewers for their feedback and suggestions. This work was supported in part by the General Project of Guangdong Provincial Natural Science Foundation (Grant No. 2026A1515010271), the General Project of Shenzhen Natural Science Foundation (Grant No. JCYJ20250604145429039), the General Project of Shenzhen Higher Education Stable Support Program (Grant No. GXWD20231130200138001), and the Science Center Program of National Natural Science Foundation of China (Grant No. 62188101).

## Impact Statement

This paper focuses on improving the robustness and energy efficiency of deep neural networks on Computing-In-Memory edge accelerators. The primary impact is to enable efficient and reliable deployment of machine learning models on resource-constrained devices.

While the work is primarily methodological, we note that enabling more efficient neural network inference could facilitate broader deployment of ML systems in practical applications. We encourage practitioners to apply responsible AI principles when deploying such systems, particularly regarding model fairness and robustness.

The work does not raise specific ethical concerns requiring further discussion.

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

# Appendix

# A. Theoretical Analysis of DWP Algorithm

In this section, we will model the nature of DWP by systematically suppressing local sharpness and guiding the optimization process towards a flatter manifold, thereby empirically demonstrating that DWP can improve the robustness of the model.

## A.1. Modeling and Variable Definition

In Computing-In-Memory (CIM) hardware, the perturbation generated by conductance deviation is proportional to the magnitude of the original parameters. We use multiplicative Gaussian noise to simulate conductance deviation. Let $\Theta \in \mathbb{R}^d$ denote the model's original parameters. We define the external pathogen perturbation as multiplicative Gaussian noise $\eta_i$. Let $\mathbf{R} \sim \mathcal{N}(0, \mathbf{I})$ be a $d$-dimensional standard Gaussian random variable. The perturbation factor for the $i$-th infection pathway is:

$$\eta_i = 1 + \zeta \mathbf{R}_i \qquad \text{(A.1)}$$

where $\zeta$ is the pathogen virulence (mismatch level) preset during training. After perturbation, the diseased weight $\Theta_i^*$ is expressed as the Hadamard product of $\Theta$ and $\eta_i$:

$$\Theta_i^* = \Theta \odot \eta_i = \Theta \odot (1 + \zeta \mathbf{R}_i) = \Theta + \zeta(\Theta \odot \mathbf{R}_i) \quad \text{(A.2)}$$

We define the stochastic infection vector $\mathbf{v}_i$ as:

$$\mathbf{v}_i = \zeta(\Theta \odot \mathbf{R}_i) \qquad \text{(A.3)}$$

where the $j$-th element of $\mathbf{v}_i$ is $v_{i,j} = \zeta \Theta_j R_{i,j}$.

## A.2. Second-Order Analysis

The DWP algorithm constructs defense by minimizing the original loss and weighted losses over $n$ infected pathways:

$$J(\Theta) = \mathcal{L}(\Theta) + \sum_{i=1}^{n} w_i \mathcal{L}(\Theta + \mathbf{v}_i) \qquad \text{(A.4)}$$

where $w_i$ is generated based on diversity awareness, satisfying the normalization condition $\sum_{i=1}^{n} w_i = 1$.

To analyze the impact of noise on the algorithm, we perform Taylor expansion of the perturbed term $\mathcal{L}(\Theta + \mathbf{v}_i)$ at $\Theta$:

$$\mathcal{L}(\Theta + \mathbf{v}_i) \approx \mathcal{L}(\Theta) + \nabla_\Theta \mathcal{L}^\top \mathbf{v}_i + \frac{1}{2} \mathbf{v}_i^\top \mathbf{H}(\Theta) \mathbf{v}_i \quad \text{(A.5)}$$

where $\mathbf{H} = \nabla^2 \mathcal{L}$ is the Hessian matrix.

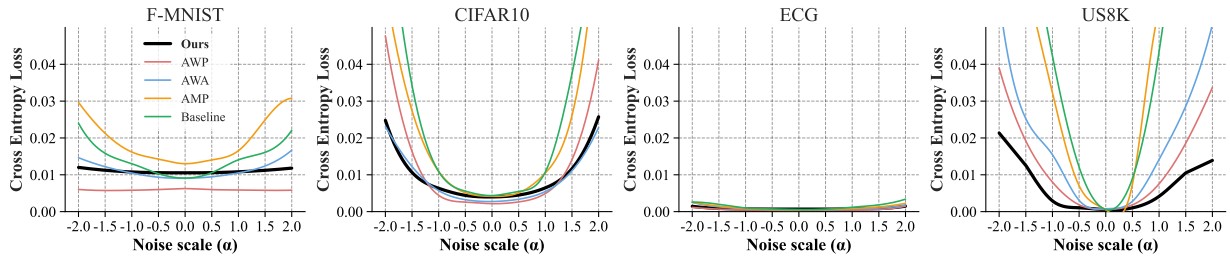

**Figure A1.** **Test loss landscape under cross-mismatch evaluation.** We use mismatch level $\zeta_1 = 0.2$ for training and $\zeta_2 = \alpha\zeta_1$ for testing. With the shift of $\alpha$, the model trained by DWP has better convergence. Our training method flattens the test weight-loss landscape.

Substituting Eq. (A.5) into Eq. (A.4):

$$J(\Theta) \approx \mathcal{L}(\Theta) + \sum_{i=1}^{n} w_i \left[ \mathcal{L}(\Theta) + \nabla_\Theta \mathcal{L}^\top \mathbf{v}_i + \frac{1}{2} \mathbf{v}_i^\top \mathbf{H}(\Theta) \mathbf{v}_i \right]$$
(A.6)

Utilizing $\sum w_i = 1$ and considering the stochastic property $\mathbb{E}[\mathbf{v}_i] = 0$, the gradient term $\sum w_i \nabla_\Theta \mathcal{L}^\top \mathbf{v}_i$ vanishes across multi-path aggregation, and the optimization objective simplifies to:

$$J(\Theta) \approx 2\mathcal{L}(\Theta) + \frac{1}{2} \sum_{i=1}^{n} w_i (\mathbf{v}_i^\top \mathbf{H} \mathbf{v}_i)$$
(A.7)

**Clarification on covariance interpretation.** We clarify that the diversity-aware weighting mechanism should be understood as a curvature-based mechanistic interpretation rather than a strict proof of covariance diagonalization. Under small perturbations, the perturbed logits can be locally linearized as $h_i \approx h + J v_i$, where $J$ is the logit Jacobian and $v_i$ denotes a weight perturbation. A second-order expansion of the cosine-consensus score shows that its dominant dependence on $v_i$ can be approximated by a quadratic form $q_i = -v_i^\top G v_i$, where $G \succeq 0$ is determined by the local logit Jacobian.

Since the diversity-aware weight $\omega_i$ is a monotone transformation of the consensus score $S_i$, the weighting rule can be interpreted as assigning larger weights to perturbation directions that induce more consistent prediction responses. If $G$ is locally diagonalizable, the weighted perturbation covariance is encouraged to concentrate along the curvature basis induced by $G$, suppressing off-dominant interactions in expectation. This does not imply pointwise diagonalization in the original parameter coordinates.

Substituting the weighted perturbation distribution into the second-order loss expansion gives

$$\mathbb{E}_v[\mathcal{L}(\Theta + v)] \approx \mathcal{L}(\Theta) + \frac{1}{2} \mathrm{Tr}\left( \mathbb{E}[vv^\top] H(\Theta) \right),$$

where $H(\Theta)$ is the local Hessian. Therefore, DWP can be interpreted as reducing sensitivity to perturbation directions that induce unstable loss variations, thereby encouraging a flatter local geometry.

## A.3. Mathematical Essence of Diversity Selection

This is the core step to demonstrate the superiority of DWP. Using the matrix property $\mathbf{x}^\top \mathbf{A} \mathbf{x} = \mathrm{Tr}(\mathbf{x}\mathbf{x}^\top \mathbf{A})$, we rewrite Eq. (A.7):

$$J(\Theta) \approx 2\mathcal{L}(\Theta) + \frac{1}{2} \mathrm{Tr}\left( \left( \sum_{i=1}^{n} w_i \mathbf{v}_i \mathbf{v}_i^\top \right) \mathbf{H} \right)$$
(A.8)

Define the perturbation covariance matrix:

$$\mathbf{M} = \sum_{i=1}^{n} w_i \mathbf{v}_i \mathbf{v}_i^\top$$
(A.9)

Examining the $(j, k)$-th element of $\mathbf{M}$:

$$
\begin{aligned}
M_{jk} &= \sum_{i=1}^{n} w_i v_{i,j} v_{i,k} \\
&= \sum_{i=1}^{n} w_i (\zeta \Theta_j R_{i,j})(\zeta \Theta_k R_{i,k}) \\
&= \zeta^2 \Theta_j \Theta_k \sum_{i=1}^{n} w_i R_{i,j} R_{i,k}.
\end{aligned}
$$
(A.10)

We consider two cases:

**Diagonal elements** $(j = k)$:

$$M_{jj}^{(t)} = \zeta^2 (\Theta_j^{(t)})^2 \sum_{i=1}^{n} w_i^{(t)} R_{i,j}^2$$
(A.11)

Since $R_{i,j} \sim \mathcal{N}(0,1)$ and $\sum w_i = 1$, by the law of large numbers and expectation properties, $\sum w_i R_{i,j}^2 \approx 1$, thus:

$$M_{jj}^{(t)} \to \zeta^2 (\Theta_j^{(t)})^2 \quad \text{(Energy term)}$$
(A.12)

**Off-diagonal elements** $(j \neq k)$:

$$M_{jk}^{(t)} = \zeta^2 \Theta_j^{(t)} \Theta_k^{(t)} \sum_{i=1}^{n} w_i^{(t)} R_{i,j} R_{i,k}$$
(A.13)

In standard random sampling (Martens et al., 2010), $\sum R_{i,j} R_{i,k}$ is often non-zero due to finite $n$. However, DWP through the cooperation of $w_i$ and $\Theta^{(t)}$ enforces the off-diagonal correlation to be suppressed in regions of extremely low flatness, such that:

$$\sum_{i=1}^{n} w_i^{(t)} R_{i,j} R_{i,k} \to 0 \qquad (A.14)$$

Consequently, $\mathbf{M}$ approaches an isotropic diagonal form:

$$\mathbf{M} \approx \zeta^2 \mathrm{diag}(\Theta_1^2, \Theta_2^2, \ldots, \Theta_d^2) = \zeta^2 \mathrm{diag}(\Theta^2) \quad (A.15)$$

### A.4. Implicit Hessian Trace Regularization

Substituting Eq. (A.15) into Eq. (A.8):

$$J(\Theta) \approx 2\mathcal{L}(\Theta) + \frac{\zeta^2}{2} \mathrm{Tr}(\mathrm{diag}(\Theta^2)\mathbf{H}) \qquad (A.16)$$

Expanding the trace operation:

$$J(\Theta) \approx 2\mathcal{L}(\Theta) + \frac{\zeta^2}{2} \sum_{j=1}^{d} \Theta_j^2 \frac{\partial^2 \mathcal{L}}{\partial \Theta_j^2} \qquad (A.17)$$

Eq. (A.17) demonstrates that the DWP algorithm systematically suppresses the curvature across all feature dimensions during training (particularly the dimension corresponding to the largest eigenvalue). This enforces the model to converge to full-dimensional diagonal curvature suppression, fundamentally eliminating the "sharp crevices" from which hardware drift could escape.

### A.5. Quantitative Verification of Robustness Enhancement

To further demonstrate the effectiveness of our algorithm in improving robustness, we analyze Figure A1, the weight-loss landscape. Define the sensitivity coefficient $k = \frac{1}{2}\zeta^2 \mathbf{v}^\top \mathbf{H} \mathbf{v}$ (Zhang et al., 2023). When subjected to perturbation of strength $\alpha$ during testing, the loss increase $\Delta\mathcal{L}$ is:

$$\Delta\mathcal{L}(\alpha) = \mathcal{L}(\Theta + \alpha\mathbf{v}) - \mathcal{L}(\Theta) \approx k \cdot \alpha^2 \qquad (A.18)$$

Based on the experimental results of CIFAR10, by substituting them into the calculations, we can obtain:

- **Baseline Model**: Due to sharp loss surface, $k_{\text{base}} \approx 0.0026$

- **DWP (Ours)**: Due to strong suppression of Hessian trace, $k_{\text{ours}} \approx 0.0005$

$$\frac{\Delta\mathcal{L}_{\text{ours}}(\alpha)}{\Delta\mathcal{L}_{\text{base}}(\alpha)} = \frac{k_{\text{ours}} \cdot \alpha^2}{k_{\text{base}} \cdot \alpha^2} \approx \frac{0.0024541496}{0.0063822862} \approx 0.3846 \qquad (A.19)$$

The calculated ratio of approximately 0.3846 indicates that the curvature of the weight loss landscape for DWP is only about 38.5% of that of the baseline. This quantitative reduction in curvature demonstrates that DWP effectively flattens the loss landscape. Consequently, the model becomes significantly less sensitive to parameter perturbations, ensuring stable performance even under high mismatch levels.

## B. CIM Simulation Setup and Noise Modeling

CIM hardware inherently suffers from intrinsic device non-idealities (Joshi et al., 2020), such as conductance drift, which can degrade model performance (Nandakumar et al., 2019). This interference tends to increase over time. To test whether the DWP-trained model can maintain stable operation in CIM hardware over an extended period, we built a CIM simulator, mimicking AWA (Büchel et al., 2022), and deployed the model on it for a year to test performance changes.

### B.1. Programming Noise

When weight matrix $W$ is programmed into PCM devices, target conductances $G_T$ (ranging from zero to $G_{\max} = 25\mu S$) are applied. However, the programming process is inherently imprecise and introduces noise:

$$G_P = G_T + \mathcal{N}(0, \sigma_P) \qquad (B.20)$$

The programming noise standard deviation is conductance-dependent and modeled as:

$$\sigma_P = \max\left(-1.1731 G_T^2 + 1.9650 G_T + 0.2635, 0.0\right) \qquad (B.21)$$

This model captures the empirical observation that programming noise is particularly significant for devices with lower conductance values.

### B.2. Drift Noise

After programming, PCM devices undergo temporal drift. The conductance evolves according to a power-law model:

$$G_D(t) = G_P \left(\frac{t}{t_c}\right)^{-\nu} \qquad (B.22)$$

where $t$ is the inference time, $t_c = 0.1$ s is the reference time point immediately after programming, and $\nu$ is the

drift coefficient. The drift coefficient varies across devices according to:

$$\nu \sim \mathcal{N}(\mu_\nu, \sigma_\nu), \quad \mu_\nu = 0.081, \quad \sigma_\nu = 0.015 \quad \text{(B.23)}$$

The non-uniform nature of drift across devices makes compensation challenging. Although Global Drift Compensation (GDC) can partially mitigate drift by scaling MVM outputs, the device-to-device variation still causes significant performance degradation over time.

### B.3. Read Noise

During inference, PCM devices experience both 1/f noise and telegraph noise. Read noise is modeled using a time-dependent Gaussian distribution:

$$G_R \sim \mathcal{N}(G_D, \sigma_{nG}(t)) \quad \text{(B.24)}$$

where the read noise standard deviation is:

$$\sigma_{nG}(t) = G_D(t) \cdot Q \cdot \sqrt{\log\left(\frac{t + t_r}{t_r}\right)} \quad \text{(B.25)}$$

with $Q = \min\left(\frac{0.0088}{G_T^{0.65}}, 0.2\right)$ and $t_r = 250$ ns.

### B.4. Simulator–Hardware Alignment Experiment

To further examine the correspondence between the CIM simulator and the physical low-precision CIM platform, we conduct a matched deployment comparison under the same model and inference configuration. This experiment is intended to provide empirical evidence for whether the simulator can reasonably capture the main deployment-time behavior of the physical hardware platform, rather than to claim strict device-level calibration.

Specifically, we deploy the same DWP-trained model for FashionMNIST inference on both the CIM simulator and the physical low-precision CIM platform. The simulator is configured to match the low-bit inference setting of the physical platform, and both evaluations follow the same inference protocol. As shown in Table B1, the CIM simulator achieves an accuracy of $84.3 \pm 1.11\%$, while the physical CIM platform achieves $82.93 \pm 1.41\%$. The comparable inference accuracy between the two platforms provides empirical support for the reasonableness of the simulator under our experimental setting.

We note that real CIM hardware involves multiple sources of non-ideality, including device mismatch, programming errors, conductance drift, read noise, and peripheral circuit

*Table B1.* Matched deployment comparison between the CIM simulator and the physical CIM platform.

| Setting | Accuracy (%) |
|---|---|
| CIM Simulator | $84.3 \pm 1.11$ |
| Physical CIM Platform | $82.93 \pm 1.41$ |

effects. Therefore, precisely reproducing the full device-level noise distribution in simulation is challenging. Instead of serving as a fully calibrated device-level simulator, our CIM simulator is used as a controllable and reproducible environment for evaluating CIM-style weight perturbation and drift. The matched deployment results indicate that, under the considered setting, the simulator can reflect the main inference behavior of the physical CIM platform to a reasonable extent.

## C. Training Costs

To comprehensively evaluate the efficiency of our proposed method, we measured the average training time per epoch across different training schemes. The results are summarized in Table C2. As observed, baseline and Forward Noise are the most efficient (10s/epoch). Adversarial training variants like AWP and AMP introduce a moderate overhead (16-17s/epoch) due to the generation of perturbations. DWP requires approximately 32s per epoch. This increase in training time (3.2×relative to the baseline) is primarily attributed to the computation required for the diversity-guided noise selection mechanism and the multiple forward passes involved.

However, it is important to note that this overhead remains significantly lower than computationally intensive methods such as AWA, which consumes 67s per epoch. Given the substantial improvements in robust generalization demonstrated in the main text, we consider the additional computational cost of our method to be a worthwhile trade-off, offering a balanced compromise between training efficiency and model robustness.

*Table C2.* Computational Cost Analysis.

| Method | Time / Epoch (s) | Relative Cost |
|---|---|---|
| baseline | 10 | 1.0× |
| Forward Noise | 10 | 1.0× |
| AWP | 16 | 1.6× |
| AMP | 17 | 1.7× |
| **Ours (DWP)** | **32** | **3.2×** |
| AWA | 67 | 6.7× |

# D. Network Quantity Selection

Theoretically, the number of noise-perturbed networks **n** determines the resolution of the stochastic manifold approximation in the DWP framework. A larger **n** enables the affinity-derived weights to more accurately reflect the consensus of diverse pathogenic directions, ensuring the training process accounts for a near-complete spectrum of noise scenarios. However, this enhanced representativeness incurs a linear increase in computational overhead.

To identify an optimal balance between computational tractability and model fidelity, we conducted a sensitivity analysis on the **CIFAR10** dataset using a **ResNet** architecture, with results summarized in Table D3. Our empirical data reveals a clear point of diminishing returns: as **n** increases from 5 to 10, the mean test accuracy improves significantly from 60.01% to 65.61%, accompanied by a reduction in standard deviation. However, further escalating **n** to 20 yields only a marginal gain of 0.82% in accuracy, while the training time per epoch nearly doubles from 32.32$s$ to 60.46$s$. Consequently, we selected **n=10** as the standard configuration for all primary experiments, as it offers the best compromise between robust generalization and training efficiency.

*Table D3.* Effect of perturbation sample size $n$ on accuracy, stability, and training time.

| $n$ | Acc.(%) | Std.(%) | Time (s/epoch) |
|---|---|---|---|
| 5 | 60.01 | 13.45 | 20.57 |
| 10 | 65.61 | 12.48 | 32.32 |
| 20 | 66.43 | 10.08 | 60.46 |

# E. Introduction to Datasets

We evaluate our method on multiple tasks across different modalities:

- **Fashion-MNIST (F-MNIST):** clothing image classification on 10 classes (Xiao et al., 2017);
- **CIFAR-10:** colour image classification on 10 classes (Kingma & Ba, 2017);
- **Labeled Faces in the Wild (LFW):** The face recognition dataset uses 34 categories. (Zhang & Deng, 2016);
- **ECG:** electrocardiogram anomaly detection on 4 classes (Bauer et al., 2019);
- **UrbanSound8K (US8K):** urban sound classification on 10 environmental sound classes (Salamon et al., 2014);
- **iFLYTEK 119:** large-scale Chinese long-text classification on 119 application-domain classes (Xu et al., 2020).

# F. Introduction to Experimental Platform

## F.1. Main Experimental Platform

Experiments in this work are primarily conducted on an NVIDIA L40 GPU. Model training and evaluation are implemented using this platform to ensure consistent computational performance and fair comparison across methods. The L40 GPU provides sufficient memory capacity and compute throughput to support large-scale training, repeated robustness evaluations, and extensive ablation studies reported in this paper.

## F.2. CIM Low-precision Platform

To enable deployment on memristive Compute-in-Memory (CIM) hardware, trained neural networks are systematically mapped onto crossbar arrays using a differential conductance representation. In our platform, we adopt a **2T2R** (two-transistor–two-resistor) architecture (Tan et al., 2020), where each synaptic weight is encoded by a pair of nonvolatile memory devices to represent positive and negative components, thereby supporting signed weights while improving linearity and robustness to device variations. Prior to mapping, network weights are clipped and linearly scaled to match the allowable conductance range of the devices (Haensch, 2024). The resulting conductance matrices are then programmed into the crossbar arrays, with large weight matrices partitioned into fixed-size tiles to accommodate array dimension constraints. During inference, input activations are converted to voltages and applied to word lines, while the corresponding matrix–vector multiplication is performed in situ through Ohm's and Kirchhoff's laws, and the accumulated currents are sensed and digitized. This mapping pipeline enables efficient execution of neural network operations while faithfully capturing the hardware characteristics and non-idealities (Zhang et al., 2024) of the underlying **2T2R CIM Platform**.

**Discussion on cross-CIM generalization.** DWP is not tied to a specific 2T2R mapping rule, because it operates at the neural-network parameter level by injecting stochastic weight perturbations during training. We use multiplicative Gaussian noise as a low-cost approximation of hardware-induced mismatch, noise, and drift, following common parameter-noise modeling practice (Büchel et al., 2022). While platform-specific device-level noise models may further improve deployment fidelity, they would also reduce the usability and generalizability of the method. Therefore, we focus on whether a general stochastic perturbation model can improve robustness to representative CIM-style non-idealities. Memristive CIM is used for physical deployment because it is a representative and relatively mature CIM technology with practical value for efficient in-situ matrix–vector multiplication (Huang et al., 2024).

# G. Additional Large-Scale Evaluation

To further address the concern regarding experimental scale, we additionally evaluate DWP on a larger-scale dataset and

*Table G4.* Additional large-scale experiment on Qwen3-1.7B for the iFLYTEK 119-class classification task. Clean Acc. and Noisy Acc. denote validation accuracy under clean and noisy evaluation, respectively.

| Noise Level | Method | Clean Acc. (%) | Noisy Acc. (%) |
|---|---|---|---|
| 0.2 | Baseline | 58.06 | $21.92 \pm 2.85$ |
| 0.2 | Ours | 53.56 | $\mathbf{36.99 \pm 2.21}$ |
| 0.4 | Baseline | 58.06 | $1.26 \pm 0.63$ |
| 0.4 | Ours | 48.17 | $\mathbf{8.93 \pm 3.99}$ |

*Table H5.* Sensitivity of DWP to different training mismatch levels on CIFAR-10. All models are tested under the same mismatch level $\zeta = 0.3$.

| Training Mismatch Level | Test Acc. (%) |
|---|---|
| 0.1 | $70.33 \pm 4.09$ |
| 0.2 | $80.24 \pm 3.59$ |
| 0.3 | $\mathbf{87.54 \pm 3.09}$ |
| 0.4 | $82.69 \pm 3.22$ |
| 0.5 | $75.59 \pm 3.88$ |

model setting. Specifically, we conduct experiments on the iFLYTEK 119-class classification task using Qwen3-1.7B. This supplementary experiment aims to examine whether the proposed weight-perturbation-based robust training strategy remains effective beyond the small- and medium-scale benchmarks used in the main evaluation.

As shown in Table G4, DWP consistently improves noisy accuracy over the baseline under different noise levels. When the noise level is 0.2, DWP improves noisy accuracy from 21.92% to 36.99%. Under a stronger noise level of 0.4, the baseline accuracy drops sharply to 1.26%, while DWP still achieves 8.93% noisy accuracy. Although robust training slightly reduces clean accuracy in the noise-free setting, the substantial improvement under noisy evaluation demonstrates that DWP remains effective in a larger-scale dataset and model configuration.

## H. Hyperparameter Sensitivity Analysis

We further analyze the sensitivity of DWP to key hyperparameters, including the training perturbation level, the number of noise realizations, the normalization parameter, and the learning rate. Among these factors, the training perturbation level is particularly important because it determines the noise distribution observed by the model during robust training.

To examine its effect, we train CIFAR-10 models under different training mismatch levels and evaluate all models under the same test-time mismatch level $\zeta = 0.3$. As shown in Table H5, the model trained with mismatch level $\zeta = 0.3$ achieves the best performance, reaching 87.54% accuracy. When the training mismatch level is smaller than the test-

time perturbation, the model does not experience sufficiently strong perturbations during training and therefore shows weaker robustness. In contrast, when the training mismatch level is overly large, clean feature learning can be disturbed, leading to degraded performance. These results indicate that DWP is affected by the training perturbation level, and robustness is generally stronger when the training perturbation better matches the deployment-time perturbation.

For other hyperparameters, the number of noise realizations controls the diversity of perturbed forward propagations, the normalization parameter stabilizes the diversity-aware weighting, and the learning rate affects the overall optimization stability. In our experiments, we keep these hyperparameters fixed across compared methods unless otherwise specified, and we observe that the main robustness improvement of DWP does not rely on a single carefully tuned configuration.

## Code and Additional Materials

The core implementation of DWP, experimental scripts, and additional materials for reproducibility are available at: `https://github.com/Burgerhai/dwp`.

