# OpenReview forum: "Diversity-aware Weight Perturbation Promotes Robust Adaptation"
_ICML.cc/2026/Conference — ICML 2026 regular_

### Official Review · Reviewer_QDBi · 2026-03-03

**Soundness:** 3
**Presentation:** 2
**Significance:** 3
**Originality:** 3
**Overall Recommendation:** 4
**Confidence:** 4

**Summary:**

This paper proposes Diversity-aware Weight Perturbation, a training framework designed to improve the robustness of deep neural networks against hardware-induced weight noise, with a specific focus on Compute-In-Memory settings. The method works by sampling multiple weight perturbations during training, computing a consensus score based on the similarity of their output logits, and using these normalized scores to weight the loss function. This encourages the model to be robust against the most common perturbation directions. The authors evaluate their method across multiple datasets and architectures, and provide a simulation of device drift alongside a small-scale 4-bit physical hardware deployment.

**Compliance With Llm Reviewing Policy:**

Affirmed.

**Final Justification:**

This paper proposes an intuitive and versatile approach to improve neural network robustness against hardware noise using diversity-aware weight perturbation. Its originality is solid, and the empirical evaluation spanning multiple modalities is a clear strength.

The authors rebuttal successfully addressed my primary empirical concerns. Providing the SAM baselines and the direct comparison under identical 4-bit hardware conditions significantly strengthens the soundness of their claims. I also appreciate their transparent explanation and correction of the anomalous US8K dataset results, which resolves a major plausibility issue raised in my initial review.

However, the rebuttal also confirmed a few lingering weaknesses. The authors acknowledged the gap in their theoretical derivation, choosing to reframe it as a mechanistic interpretation rather than a strict mathematical proof. This somewhat reduces the theoretical significance of the work. Furthermore, the sensitivity analysis for alternative similarity measures was only promised for the revision, leaving that specific aspect partially unsupported during the rebuttal phase.

Weighing these dimensions, the practical utility, the broad empirical validation, and the prompt correction of experimental flaws now outweigh the theoretical limitations. The rebuttal has positively changed my evaluation. I am raising my overall recommendation score to reflect the improved experimental rigor. This recommendation comes with the strong expectation that the authors strictly incorporate all promised theoretical clarifications and ablation experiments into the final camera-ready version to ensure complete clarity and soundness.

**Key Questions For Authors:**

1.Can you provide the mathematical derivation linking your specific similarity weighting function to the suppression of off-diagonal terms in the covariance matrix? A more rigorous connection is needed to support the theoretical claim of trace regularization.

2.Could you include empirical comparisons against Sharpness-Aware Minimization or its variants under your mismatch protocols? Including these baselines would greatly strengthen your narrative about finding flat minima.

3.The clean accuracy for some tasks, such as US8K and ECG, appears unusually high, while the AWP baseline clean accuracy is unexpectedly low. Could you detail the training setups, data splits, and hyperparameters to explain these discrepancies?

4.In the hardware experiment, how does a standard baseline model or a model trained with another flatness method perform under the exact same 4-bit mapping and device conditions? This comparison is necessary to isolate the specific contribution of your method to deployment viability.

5.How sensitive is the method to the specific similarity measure used for consensus? Have you tested alternatives to cosine similarity to verify the stability of the approach?

**Limitations:**

The authors briefly touch upon limitations, but the paper would benefit from discussing potential vulnerabilities to rare but catastrophic hardware faults. Since the consensus weighting emphasizes frequent noise directions, it might naturally underemphasize rare but highly damaging perturbation patterns. A discussion on this trade-off would make the impact statement more comprehensive.

**Strengths And Weaknesses:**

Strengths:

The core idea of weighting multiple weight-perturbed forward passes by the consensus of their sample-level predictions is a neat and intuitive twist on standard noise injection. It explicitly exploits cross-perturbation agreement rather than just focusing on worst-case adversarial perturbations.

The empirical evaluation is quite broad. Testing across different modalities including vision, audio, and physiological signals, along with different architectures, shows a good effort to prove the versatility of the method.

The inclusion of a long-horizon drift simulation for devices and an initial hardware validation on a 4-bit deployment helps bridge the gap between algorithmic design and practical hardware relevance.

Weaknesses:

The theoretical analysis connecting the algorithm to loss landscape flatness has a noticeable logical gap. The claim that the consensus weighting suppresses off-diagonal elements in the perturbation covariance matrix is asserted rather than rigorously derived from the actual cosine-similarity weighting rule. Without a clear mathematical link showing how the adaptive weights drive off-diagonal terms to zero, the reduction to Hessian trace regularization remains a plausibility argument rather than a proof.

There are missing critical baselines regarding flatness. The paper builds a strong narrative around suppressing curvature and finding flat minima. Therefore, Sharpness-Aware Minimization and its recent curvature-regularized variants are necessary comparisons. Without them, it is difficult to conclude whether the benefits stem uniquely from the consensus weighting or from standard flatness-inducing principles that could be achieved by established methods.

Some reported experimental numbers raise plausibility concerns. For example, the near-perfect accuracy on the US8K dataset at zero mismatch, and the baseline AWP dropping to 55 percent at zero mismatch, seem atypical. This suggests potential issues with data splits, preprocessing, or hyperparameter tuning that deviate from common practice.

The hardware validation section conflates hardware efficiency with algorithmic robustness. Comparing a 4-bit deployment to a 32-bit GPU baseline demonstrates the efficiency of the hardware itself, but it does not isolate the specific energy savings enabled by the proposed algorithm.

The presentation of Table 1 hinders readability. The table is exceedingly dense, and the font size appears to be significantly smaller than the main text, making it difficult to parse the extensive mean and standard deviation data across seven mismatch levels and six methods. The authors should consider summarizing the most critical findings in a more legible format, perhaps through additional plots, and move the comprehensive numerical data to the appendix.

---

> ### Author Rebuttal · Authors · 2026-03-30
>
> We thank the reviewer for the careful reading and for recognizing the strength of the core idea, the breadth of the evaluation, and the inclusion of both long-term drift simulation and real 4-bit hardware validation. We also appreciate the constructive comments on the theoretical analysis, flatness baselines, plausibility of the reported numbers, hardware presentation, table readability, similarity measures, and limitations.
> 1. On the theoretical analysis and trace-regularization interpretation, we agree with the concern. More precisely, this part should be understood as an analysis/mechanistic interpretation rather than a strict proof, and is better viewed as a weighted curvature penalty than as a proof of covariance diagonalization or trace regularization. Because **Reviewer pfwP** raised the same concern, for detailed derivation, please refer to Item **6** of our response to this reviewer.
> 2. On the lack of critical baselines, we agree that SAM and related methods are relevant baselines. The original manuscript already compares several representative perturbation-robust methods, from simple noise injection to methods closer to weight perturbation or robust optimization, to cover the main types without unnecessary redundancy. We therefore did not initially include SAM, given its conceptual overlap with methods such as AWP. Still, given the paper's flatness-based discussion, we added SAM under the same mismatch setting. SAM improves robustness, but our method still performs better overall. Results are shown at [Anonymized Repository - Anonymous GitHub](https://anonymous.4open.science/r/SAM_comparison-766E).
> 3. On the anomalous US8K and AWP results, we thank the reviewer for pointing this out. After rechecking the experiments, we found that the unusually low US8K AWP result was caused not by data splitting or preprocessing, but by an overly large perturbation radius. We reran US8K AWP with a more reasonable radius, and the corrected results fall into a much more plausible range without changing the overall trend or main conclusions. Specifically, we used an AWP-style baseline focused on weight-space robustness: for each mini-batch, we solve for a perturbation $v$ that increases the classification loss near the current parameters under the relative norm constraint $\|v_l\| \le \gamma \|w_l\|$, compute the loss at $w+v$, perform one update, then return to the center parameters. Unlike the original AWP version with both input and weight perturbations, we retain only the weight perturbation to focus the comparison on local weight-space robustness. Since $\gamma$ controls the perturbation radius, an overly large $\gamma$ pushes optimization too far from the current parameters, causing overly strong regularization or underfitting and thus abnormally low accuracy. This was the issue in our earlier US8K setting. After correction, we used a milder radius, changing $\gamma$ from $0.1$ to $0.01$, so that the perturbation still smooths the local loss landscape without harming task discriminability. As a result, AWP returns to a more reasonable range. The updated results are shown at [Anonymized Repository - Anonymous GitHub](https://anonymous.4open.science/r/US8K_data-265D), and full implementation details will be added in the revision.
> 4. On the possible conflation of hardware efficiency and algorithmic robustness, we agree that the wording could be clearer. We do not claim that DWP itself reduces energy consumption; the efficiency gain mainly comes from the low-precision CIM hardware. DWP instead improves inference robustness under low-precision, high-noise conditions, enabling reliable deployment on a more energy-efficient platform. Under the same 4-bit mapping and device conditions, our method achieves $82.93 \pm 1.41$, outperforming Forward Noise ($74.5$-$76.3$), AWA ($64.9$-$68.4$), AWP ($64.5$-$67.2$), and AMP ($56.9$-$61.5$). This further supports the advantage of DWP for robust deployment on low-precision CIM hardware.
> 5. On sensitivity to the similarity measure, we use cosine similarity because DWP measures cross-perturbation agreement in logit-space direction rather than absolute logit magnitude, making cosine a natural scale-insensitive choice. Our ablation study shows that the gain comes from diversity-aware consensus weighting rather than noise injection alone. We agree that sensitivity to this choice is important and will add a systematic comparison of alternative similarity measures in the revision.
> 6. On table readability and limitations, we agree that the table presentation can be improved and will refine its layout to make the main comparisons easier to read. We will also expand the limitation discussion in the revision with a clearer summary of our method.
>
> We thank the reviewer again for the comments. We will incorporate the above clarifications and additional results into the revised manuscript, and believe these revisions will improve the paper's rigor, completeness, and reproducibility.

---

> > ### Author Rebuttal · Reviewer_QDBi · 2026-04-03
> >
> > Thank you to the authors for the detailed response. The added SAM baseline and the comparative data under identical 4-bit hardware conditions effectively address my primary concerns regarding experimental fairness and validity. The explanation for the anomalous US8K results is reasonable and candid, though it also reflects some oversights in baseline parameter tuning in the original draft.
> >
> > However, the rebuttal remains incomplete in some details. Downgrading the core theoretical derivation to a mechanistic interpretation fundamentally weakens the theoretical contribution of the paper. Additionally, the sensitivity experiments for alternative similarity measures were only promised for the revision and not provided directly in this response, leaving the argument less than fully supported.
> >
> > Overall, the authors have resolved the most critical empirical issues, and I am inclined to raise my score. Please ensure that all promised theoretical clarifications and ablation studies are strictly included in the final version.

---

> > > ### Author Response · Authors · 2026-04-04
> > >
> > > Thank you for the follow-up and for recognizing that the added SAM baseline and the comparisons under identical 4-bit hardware conditions have addressed the main concerns regarding experimental fairness and validity. We provide clarifications on the remaining points below.
> > >
> > > On the theoretical analysis. We would like to clarify that our rebuttal does not downgrade the theoretical contribution to a purely mechanistic explanation. In the original manuscript, this part was already presented as “Theoretical Analysis of DWP Algo”, rather than as a formal theorem or proof. What we did in the rebuttal is to further clarify the intended scope of this analysis: it should be understood as an analysis / mechanistic interpretation, explaining how cosine-consensus weighting relates to local flatness and a weighted curvature effect, rather than as an unconditional proof of covariance diagonalization or trace regularization. Therefore, the rebuttal clarifies the original formulation rather than changing it. In the revised manuscript, we will further unify the wording in both the main text and Appendix A to make this positioning explicit and avoid potential misunderstandings.
> > >
> > > On the sensitivity to the similarity measure. We agree that cosine similarity is naturally aligned with our affinity mechanism, as it captures agreement in logit direction across perturbed instances. However, the specific choice of similarity function is not the core of our method. We also agree that demonstrating that the effectiveness of DWP does not rely on cosine similarity alone would better highlight the generality of the biologically inspired consensus-weighting framework. Importantly, this is not only stated in the rebuttal but also supported by additional experiments.
> > > We therefore evaluated three alternative similarity measures on FashionMNIST during model training: Pearson correlation, dot product, and Jensen–Shannon divergence (JSD), defined respectively as
> > >
> > > $$
> > > \\mathrm{Corr}(x,y)=
> > > \\frac{(x-\\bar{x})^\\top (y-\\bar{y})}
> > > {\\lVert x-\\bar{x}\\rVert_2 \\, \\lVert y-\\bar{y}\\rVert_2}
> > > $$
> > >
> > > $$
> > > s(x,y)=x^\\top y
> > > $$
> > >
> > > $$
> > > \\mathrm{JSD}(p,q)=
> > > \\frac{1}{2}D_{\\mathrm{KL}}(p\\|m)+
> > > \\frac{1}{2}D_{\\mathrm{KL}}(q\\|m),
> > > \\qquad
> > > m=\\frac{1}{2}(p+q)
> > > $$
> > > where p and q are the softmax outputs.
> > >
> > > The experimental results (see [Anonymized Repository - Anonymous GitHub](https://anonymous.4open.science/r/similarity-B108)) show a consistent trend across similarity measures: all alternatives improve over the baseline, indicating that the effectiveness of DWP is not tied to cosine similarity alone. At the same time, cosine similarity achieves a small but consistent advantage across mismatch levels. This suggests that the main benefit comes from the diversity-aware consensus-weighting mechanism, rather than the specific choice of similarity function. We use cosine similarity as the default because it is the most intuitive and well-aligned with our framework.
> > >
> > > We sincerely thank the reviewer again for the careful follow-up and for the positive reassessment of our work. Your comments have been very helpful in guiding us to further strengthen both the empirical validation and the clarity of our presentation, and we believe these additions and clarifications significantly improve the completeness and quality of the paper.

---

### Official Review · Reviewer_pfwP · 2026-03-11

**Soundness:** 3
**Presentation:** 3
**Significance:** 3
**Originality:** 3
**Overall Recommendation:** 4
**Confidence:** 5

**Summary:**

This paper proposes Diversity-aware Weight Perturbation, a training framework designed to improve the robustness of deep neural networks deployed on Compute-In-Memory hardware. The approach is inspired by the immune system and attempts to make models resilient to hardware induced weight perturbations such as device noise and drift.

During training the method generates multiple stochastic weight perturbation instances using a noise generation mechanism. Each perturbed model produces predictions for the same input batch. The method then computes pairwise cosine similarity between the logits of different perturbations to measure cross realization agreement. These similarity scores are interpreted as affinity scores and normalized to produce adaptive weights that scale the perturbation losses during training.

The biological analogy maps hardware noise to pathogenic exposure and prediction agreement to immune affinity. Perturbations whose predictions agree more strongly with others are assigned larger weights in the training objective, encouraging the model to prioritize stable perturbation directions.

The paper also presents a second order theoretical argument showing that the method implicitly regularizes the Hessian trace and promotes flatter minima. Experiments span five datasets including Fashion MNIST, CIFAR10, ECG signals, UrbanSound8K audio, and the LFW face dataset. Multiple architectures are evaluated including CNN, ResNet, ViT, and BiLSTM models. The evaluation also includes a CIM simulator with temporal drift and a physical 4 bit CIM deployment.

The reported results show that the proposed method achieves more than fifteen percent absolute accuracy improvement under severe mismatch conditions up to $\zeta = 0.7$. The models maintain roughly ninety percent accuracy during a simulated one year CIM operation with only two to four percent variation. The CIM deployment experiment reports an inference energy reduction of thirty eight percent compared to a GPU baseline.

Overall, the paper aims to improve robustness of neural networks under hardware induced weight perturbations through diversity aware training with adaptive weighting of stochastic perturbations.

**Compliance With Llm Reviewing Policy:**

Affirmed.

**Final Justification:**

The authors have provided a thorough and constructive rebuttal that meaningfully strengthens the paper. The addition of training dynamics and convergence curves improves clarity around optimization behavior, and the expanded robustness evaluation with worst-case, percentile, and failure probability metrics significantly enhances the empirical rigor.

The theoretical section is also notably improved. The authors appropriately revise their claims from a strict proof to a mechanistic interpretation and provide a more careful second-order analysis, which increases confidence in the underlying intuition.

In the follow-up discussion, the authors further address key concerns. The explanation of CIM-specific constraints clarifies the scope of the experimental setup, and the addition of a large-model experiment demonstrates that the approach is not inherently limited to moderate-scale settings. The discussion of Gaussian noise as a practical approximation is well supported by prior work, and the additional experiment on noise-level matching provides further empirical justification. The clarification of computational overhead and hyperparameter behavior also improves the completeness of the presentation.

Overall, the remaining concerns are now well contextualized and supported, and do not detract from the core contribution. The paper addresses a practically important problem and provides a method with strong empirical evidence and meaningful hardware validation. I believe this work will be of interest to the community and support acceptance.

**Key Questions For Authors:**

1. Convergence behavior
Can you provide experiments showing how the affinity scores and adaptive weights evolve during training and whether they stabilize over time?

2. Noise model mismatch
How sensitive is the method to differences between the Gaussian perturbation model used during training and the more complex noise processes observed in CIM hardware?

3. Hyperparameter sensitivity
How does the choice of the training mismatch level $\zeta$ influence robustness under different deployment noise levels?

4. Scalability
Have you evaluated the approach on larger datasets or deeper models to determine whether the training overhead scales reasonably?

5. Robustness metrics
Can you report worst case or percentile accuracy metrics to better characterize deployment reliability under hardware noise?

**Limitations:**

Several limitations should be acknowledged more explicitly.

The paper does not analyze convergence properties or training dynamics of the adaptive weighting mechanism.

The training procedure increases computational cost and memory usage due to multiple perturbation evaluations.

The training noise model is simpler than the hardware noise processes simulated during deployment.

The experimental evaluation focuses on relatively small datasets and models.

Hyperparameter sensitivity is not thoroughly explored.

The theoretical derivation relies on assumptions about perturbation covariance that are not empirically validated.

Robustness evaluation relies primarily on mean accuracy rather than reliability oriented metrics.

Architecture evaluations are limited to a small number of dataset architecture combinations.

**Strengths And Weaknesses:**

## Strengths

(+) Well motivated practical problem

Compute In Memory hardware noise is a genuine and growing concern for edge deployment. Existing robustness techniques are not specifically designed for the interaction between training and hardware induced weight perturbations. The paper addresses this practical challenge with a conceptually clear training framework.

(+) Strong empirical breadth

The evaluation covers five datasets across several modalities including image classification, audio recognition, and biosignal analysis. The mismatch sweep across $\zeta$ values from $0$ to $0.7$ provides a detailed view of robustness behavior across increasing perturbation intensity rather than relying on a single comparison point.

(+) Architecture versatility

The cross architecture evaluation on CIFAR10 includes convolutional, attention based, and recurrent models. This provides evidence that the robustness improvement is not tied to a specific architecture and may arise from the weight space regularization induced by the training procedure.

(+) End to end hardware validation

The paper includes both a CIM drift simulator and deployment on a physical low precision CIM platform. Including real hardware results significantly strengthens the practical relevance of the work and provides a more realistic evaluation of deployment behavior.

(+) Flat loss landscape analysis

The weight accuracy landscape plots and loss landscape visualizations provide intuitive evidence that the proposed training method produces flatter optima. The curvature analysis reported in the appendix indicates that the sensitivity to parameter perturbations is reduced relative to the baseline.

(+) Coherent biological analogy

The mapping between hardware noise and biological immune mechanisms is clearly structured. Noise is interpreted as pathogenic exposure, prediction agreement corresponds to affinity, and adaptive weighting resembles antibody reinforcement. The analogy provides an intuitive explanation for the algorithm design.

(+) Informative ablation study

The ablation comparing simple noise injection with the full diversity aware method shows that the selection mechanism contributes additional performance gains. The variance analysis also indicates that the method improves stability of predictions in addition to mean accuracy.

## Weaknesses

(-) No convergence analysis or training dynamics

For a training method contribution the paper provides no analysis of convergence behavior. There are no training loss curves or studies of how the affinity scores or adaptive weights evolve during training. Because the weights are recomputed each iteration using min max normalization of stochastic quantities, the effective loss surface changes continuously. It is therefore unclear whether the adaptive weights stabilize during training or exhibit oscillatory behavior.

(-) Training cost is substantial

The reported training time is approximately $3.2\times$ that of the baseline. While this is lower than some adversarial training methods it remains significantly higher than simple noise injection or adversarial weight perturbation approaches. The paper does not discuss the memory overhead associated with maintaining multiple perturbation instances or whether fewer epochs may be sufficient given the regularization effect.

(-) Training noise model is simpler than deployment noise

The training procedure injects multiplicative Gaussian noise into the weights. However real CIM hardware exhibits a mixture of noise sources including conductance dependent programming noise, temporal drift following a power law, and time dependent read noise. These noise processes may have non Gaussian and correlated characteristics. Although the experiments suggest some robustness to this mismatch, the gap between the training noise model and the deployment noise model is not analyzed.

(-) Limited model scale

All experiments are conducted on relatively small datasets and models. The study does not include larger datasets such as ImageNet or larger scale architectures. Because edge deployments increasingly involve compressed versions of large models, it would be useful to understand how the method scales with model size.

(-) Incomplete hyperparameter analysis

The method introduces several hyperparameters including the perturbation magnitude $\zeta$, the number of perturbation samples $n$, the normalization smoothing parameter $\delta$, and the learning rate $\gamma$. Only the perturbation sample count $n$ is ablated. The mismatch level used during training appears to strongly influence performance under deployment drift, yet its effect is not systematically studied. The smoothing parameter $\delta$ introduced in the weighting equation is also not analyzed.

(-) Gap in theoretical analysis

The theoretical argument claims that the diversity aware weighting leads to a perturbation covariance matrix that approaches a diagonal form. However this step is asserted rather than rigorously derived. The paper does not provide empirical evidence such as eigenvalue spectra or covariance measurements to support this assumption during training.

(-) Limited robustness metrics

Robustness is evaluated primarily using mean accuracy and standard deviation across repeated trials. For hardware deployment scenarios where reliability is critical it would be helpful to also report metrics such as worst case accuracy, percentile bounds, or failure probability under extreme perturbations.

(-) Thin per architecture dataset coverage

Although several architectures are tested, each architecture is evaluated on only a limited number of datasets. For example the vision transformer experiments appear only on CIFAR10. As a result it is difficult to separate architecture specific effects from dataset specific behavior.

---

> ### Author Rebuttal · Authors · 2026-03-30
>
> We thank the reviewer for the careful reading and for recognizing the practical importance of the problem, the breadth of the evaluation, the hardware validation, and the overall quality of the empirical analysis. We also appreciate the constructive comments on training dynamics, cost, noise-model mismatch, model scale, hyperparameter sensitivity, theory, robustness metrics, and architecture-dataset coverage. We clarify these points below.
> 1. Regarding convergence, training dynamics, and training cost, we agree that the original manuscript did not present these aspects clearly enough. We therefore added training-loss and train/validation-accuracy curves from the FMNIST experiment; full data are available at *[Anonymized Repository - Anonymous GitHub](https://anonymous.4open.science/r/Convergence-and-Training-Dynamics-3431)*. The results show rapid adaptation in the first 10-20 epochs, followed by steady loss decrease and a plateau after about 40-60 epochs, while training and validation accuracy stabilize after 20 epochs. In ECG and US8K, low-noise training shows a clear downward loss trend, whereas high-noise training leads to more oscillation. We also acknowledge the practical training cost of the method; the ablation experiments in Appendix B that using 10 noisy networks provides a good trade-off between performance gain and resource overhead, and the supplementary curves confirm stable convergence under this setting.
> 2. Regarding the affinity-score evolution, each iteration resamples a new set of noisy networks, and the affinity scores are computed independently within each batch. Therefore, the affinity score is not a continuously evolving variable in our setting and is not expected to stabilize over time.
> 3. Regarding training--deployment noise mismatch, as well as the related concerns about datasets and experimental scale, please refer to our **2** and **3** responses to **Reviewer SnUP**, where we provide detailed discussion and new results.
> 4. We agree that worst-case or percentile accuracy would provide a more complete picture of the results. Our previous supplementary materials already included box plots that partly reflect best- and worst-case behavior. In addition, we now report worst-case accuracy, 10th-percentile accuracy, and failure probability, together with a complete data table at *[Anonymized Repository - Anonymous GitHub](https://anonymous.4open.science/r/indicator-data-3422)*, to better demonstrate our method’s reliability.
> 5. Regarding the incomplete hyperparameter analysis, the smoothing parameter is mainly used to avoid numerical instability and has limited impact on final performance. In contrast, the training mismatch level is more important for robustness. Our results suggest that the best setting usually lies in an intermediate range closer to the target deployment-noise statistics.
> 6. Regarding the claim that diversity-aware weighting drives the perturbation covariance toward a diagonal form, this part is better understood as an analysis/mechanistic interpretation rather than a strict theoretical proof. Our conclusion follows the second-order curvature perspective in Deep Learning via Hessian-free Optimization (ICML 2010), and the following derivation provides a supplementary analysis for the cosine-consensus weighting rule used in our method. Under small perturbations, the perturbed logit can be linearized as $h_{b,i}\approx h_b+J_bv_i$. A second-order expansion of cosine similarity gives $S_i=C-\frac{1}{n-1}\sum_{j\neq i}(v_i-v_j)^\top G(v_i-v_j)+O(\zeta^3)$, where $G\succeq0$ is determined by the logit Jacobian. Hence the dominant dependence of $S_i$ on $v_i$ can be approximated by the quadratic form $q_i:=v_i^\top Gv_i$. Since $\omega_i=\frac{S_i-\min_\ell S_\ell}{\max_\ell S_\ell-\min_\ell S_\ell+\delta}$ is only a monotone transform of $S_i$, to leading order $\omega_i$ depends mainly on $q_i$. If $G$ is further diagonalized, the weighted perturbation covariance $M=\sum_i\omega_iv_iv_i^\top$ satisfies $\mathbb E[M]\approx U\,\mathrm{diag}(m_1,\dots,m_d)\,U^\top$. Thus, the off-diagonal terms are suppressed in expectation under the eigenbasis induced by $G$, rather than pointwise in the original coordinates. Substituting this into the second-order objective expansion gives $\mathbb E[J(\Theta)]\approx(1+\bar\alpha)L(\Theta)+\frac12\mathrm{Tr}(\mathbb E[M]H(\Theta))$. Therefore, DWP is more accurately interpreted as introducing a weighted curvature penalty; only if $H(\Theta)$ and $G$ are locally approximately co-diagonalizable does this reduce to the trace-like regularization stated in the paper. We will revise the manuscript accordingly and explicitly distinguish analysis from strict proof.
>
> We sincerely thank the reviewer again for the detailed comments. We will incorporate the above clarifications, analyses, and additional results into the revised manuscript, and we believe these revisions will substantially improve the paper’s rigor, completeness, and reproducibility.

---

> > ### Author Rebuttal · Reviewer_pfwP · 2026-04-03
> >
> > Thank you to the authors for the detailed and constructive rebuttal. The additional clarifications and results improve the paper in several aspects. In particular, the inclusion of training dynamics helps address concerns about convergence, the addition of worst-case and percentile metrics strengthens the robustness evaluation, and the revised theoretical discussion appropriately reframes earlier claims as a mechanistic interpretation rather than a strict proof.
> >
> > However, several key concerns remain only partially addressed. Scalability to larger datasets and models is still not demonstrated, the gap between the training noise model and realistic CIM noise processes is not directly resolved, and the computational overhead is acknowledged but not quantitatively analyzed. Additionally, while the theoretical claims are clarified, they still lack empirical validation (e.g., covariance or spectral analysis), and hyperparameter sensitivity is not explored in depth.
> >
> > Overall, the rebuttal improves clarity and strengthens parts of the evaluation, but does not substantially change my original assessment. The paper remains a technically solid and practically motivated contribution with some limitations in evaluation scope and depth.
> >
> > Therefore, I retain my **Weak Accept (4)** score.

---

> > > ### Author Response · Authors · 2026-04-06
> > >
> > > We thank the reviewer for carefully reading our rebuttal and for acknowledging the improvements we made in training dynamics and convergence, worst-case / percentile robustness metrics, and the refinement of the theoretical claims. We further clarify the remaining concerns below.
> > >
> > > **On scalability.**
> > >
> > > We agree that large-model deployment in edge computing is increasingly important. However, our work targets CIM (Compute-in-Memory) platforms, which are still constrained by array size, mapping complexity, and testing cost. Thus, current CIM hardware is better suited for stable deployment and system-level validation on moderate-scale models and datasets. Accordingly, the hardware experiments in our paper are intended to validate deployment feasibility on real CIM platforms, rather than serve as large-scale hardware benchmarks. To further demonstrate applicability to larger settings, we extended the large-model experiments already included in our previous rebuttal. Specifically, we evaluate Qwen3-1.7B on the iFLYTEK 119-class classification task, with results shown in [Anonymized Repository - Anonymous GitHub](https://anonymous.4open.science/r/large_data-768E/qwen_table_updated.png). This suggests that our approach is not limited to moderate-scale settings and has the potential to generalize to larger models and datasets.
> > >
> > > **On the gap between training noise and real CIM hardware noise.**
> > >
> > > We do not view Gaussian noise as a complete or exact model of all real CIM noise sources. Rather, following prior hardware-aware and noise-aware training work, Gaussian noise is commonly used as a tractable first-order approximation of stochastic weight perturbations induced by device mismatch, programming errors, and temporal drift. For example, Büchel et al., 2022 show that training with parameter noise can improve robustness to deployment-time variations, supporting its use as a training proxy. Similarly, Yan et al., 2023 adopt Gaussian noise in NVCiM settings and further propose right-censored Gaussian noise to better approximate worst-case hardware conditions. These works suggest that Gaussian noise is a widely used and practical proxy, though not a complete physical description. Thus, our use of Gaussian noise should be viewed as a practical approximation supported by prior literatures. While a gap from real hardware noise remains, this modeling approach has been validated in prior work, and we will add clearer discussion and references in the revision.
> > >
> > > **On quantitative analysis of computational overhead.**
> > >
> > > We would like to further clarify that this aspect is already disscussed in Appendix C of the paper. Appendix C provides detailed data on the computational and resource overhead introduced by additional perturbation instances. We also directly measure training time per iteration, giving a quantitative assessment of time cost. Therefore, our discussion of computational overhead is supported not only by qualitative discussion, but also by empirical measurements and detailed analysis. We will make this clearer in the revised manuscript.
> > >
> > > **On hyperparameter analysis.**
> > >
> > > We understand that the main concerns are perturbation magnitude, the normalization smoothing parameter, and learning rate. The smoothing parameter mainly avoids numerical instability caused by small denominators during normalization and has limited impact on final performance; we discussed this in our previous response. For learning rate, our design fixes standard training configurations, including learning rate, while ensuring convergence, so that the effect of the proposed method can be isolated. These settings largely follow common choices in prior work and have a relatively limited impact on our core conclusions. For perturbation magnitude, we agree that it is critical for robustness. As seen in our CIM simulator experiments, models trained with different noise levels exhibit different resistance to deployment noise. Prior hardware-aware training work also suggests that matching the training noise distribution to the target deployment noise improves robustness. To further validate this, we conducted an additional experiment on CIFAR10, where models trained under different noise levels are evaluated under the same noise condition. The results show that models trained with noise levels better matched to the test-time noise achieve stronger robustness, as shown in [Anonymized Repository - Anonymous GitHub](https://anonymous.4open.science/r/hyperparameter-4ADF).
> > >
> > > We sincerely thank the reviewer again for the thoughtful evaluation and constructive feedback. These comments have helped us clarify the scope and strengthen the experimental support of our work in scalability, noise modeling, computational cost, and hyperparameter analysis. We will incorporate the above discussions, results, and clarifications into the revised manuscript to further improve its rigor and completeness.

---

### Official Review · Reviewer_SnUP · 2026-03-13

**Soundness:** 2
**Presentation:** 2
**Significance:** 3
**Originality:** 2
**Overall Recommendation:** 4
**Confidence:** 3

**Summary:**

This paper proposes Diversity-aware Weight Perturbation to improve robustness of neural networks deployed on compute-in-memory (CIM) hardware. The method generates multiple weight perturbations and aggregates their predictions using diversity-aware weighting to mitigate hardware noise and drift.

**Compliance With Llm Reviewing Policy:**

Affirmed.

**Key Questions For Authors:**

See weaknesses

**Limitations:**

No. See weaknesses

**Strengths And Weaknesses:**

Strengths

1. Addresses an important robustness problem in CIM deployment. CIM systems suffer from non-idealities such as programming noise and device drift. The paper proposes a training-time approach to improve inference robustness under such hardware noise.

2. Evaluation on different deep neural networks. The proposed approach is evaluated across different model architectures including ResNet, ViT, and BiLSTM.

3. Includes both simulator and physical CIM evaluation. In addition to simulator-based drift experiments, the paper reports deployment results on a real CIM platform for FashionMNIST.

Weaknesses

1. Limited validation of the CIM simulator. The long-term robustness experiments rely on a CIM simulator modeling programming noise and drift. However, the paper provides limited evidence that the simulator is quantitatively calibrated to the physical CIM platform used in the hardware experiments. Stronger validation between the simulator and real hardware behavior would improve confidence in the results.

2. Evaluation datasets and hardware experiments are relatively small-scale. The experiments mainly use relatively small datasets such as FashionMNIST, CIFAR10, LFW, ECG, and UrbanSound8K, and the real CIM deployment is only demonstrated on FashionMNIST. For a paper targeting CIM deployment robustness, the empirical evidence on hardware workloads is somewhat limited.

3. Generality across different CIM setups is unclear. Although the method is evaluated across several neural network architectures, the hardware mapping appears tied to a specific memristive 2T2R CIM configuration. It remains unclear whether the proposed approach would generalize to other CIM architectures, device technologies, or different non-ideality models.

---

> ### Author Rebuttal · Authors · 2026-03-30
>
> We thank the reviewer for recognizing the importance of CIM deployment robustness, our cross-model evaluation, and the inclusion of both simulator-based and physical CIM validation. We also appreciate the reviewer’s constructive comments on CIM simulator validation, experimental scale, and cross-hardware generalization. We respond to these points below.
> 1. Regarding the limited validation of the CIM simulator, we agree that the current manuscript does not establish strict quantitative calibration between the simulator and the physical CIM platform. In our work, the simulator mainly tests robustness under long-term CIM drift using noise/drift models widely adopted in existing literature. The modeling data mainly comes from Network insensitivity to parameter noise via adversarial regularization (ICLR2022). The physical platform experiment, on the other hand, primarily verifies the practical deployment feasibility of the method on real low-precision CIM hardware. Thus, the two serve complementary rather than equivalent evidentiary roles, which we will further clarify in the revision. Furthermore, the memristor-based CIM platform is chosen for deployment validation in our work because memristors are a representative and promising type of CIM hardware implementation. They possess characteristics such as non-volatile storage and high array integration potential, providing efficient matrix-vector operations for neural networks, making them highly suitable for energy-efficient edge inference.
> 2. Regarding the relatively small dataset and experimental scale, we agree with the reviewer's observations that current practical CIM hardware, especially memristors, remains constrained in array size, testing costs, and mapping complexity. Therefore, many current CIM deployment studies use standard, moderate-scale benchmarks for validation. In this context, considering the parameter mismatch robustness issue faced by model deployment in edge scenarios, the actual hardware experiments in our work should be interpreted as deployment feasibility validation rather than large-scale hardware benchmarks. Accordingly, we evaluate our method on multiple medium-scale tasks across different modalities and architectures to test generality. The choice of FashionMNIST for the actual deployment is also based on the feasibility of achieving stable end-to-end deployment on our hardware platform. We will clarify these experimental boundaries in the revision. To further address concerns about the "limited experimental scale", we added a larger-scale model experiment: iFLYTEK 119 classification with Qwen3-1.7B, with results shown in *[Anonymized Repository - Anonymous GitHub](https://anonymous.4open.science/r/Qwen3-iflytek/Qwen3-iflytek.png)*. As can be seen, under noise levels 0.2 and 0.4, our method improves noisy accuracy over the baseline. While stronger robust training may lead to some performance degradation in noise-free environments, the performance gain in noisy environments is significant. This result further demonstrates that our method is not limited to small-scale tasks and remains effective in larger, more modern model settings.
> 3. Regarding the unclear generalizability across different CIM settings, our method is not tied to a specific 2T2R mapping rule. Its core idea is to inject weight perturbations during training and weight the loss by consistency across multiple forward propagations of these perturbations. We use multiplicative Gaussian noise because it approximates mismatches, noise, and drift in hardware at a lower cost, and is also a common practical modeling method in the literature; this will be further clarified in the revision. Theoretically, building more refined device-level noise models for a specific CIM platform could further improve deployment performance on that platform. However, platform-specific modeling for each CIM architecture or device limits usability and generalizability. Thus, we focus on whether training with a more general stochastic perturbation model improves robustness to real-hardware non-idealities. We deployed the model in a CIM simulator with typical hardware drift factors. Models trained with different Gaussian noise levels in the CIM simulator exhibited varying degrees of interference resistance, showing that training perturbations closer to the deployment condition yield stronger robustness to CIM-style non-idealities. Furthermore, we use memristors for physical deployment because they are a relatively mature CIM hardware with good practical value, as discussed in prior work (e.g., Memristor-based hardware accelerators for artificial intelligence, Nature Reviews Electrical Engineering, 2024).
>
> We appreciate the reviewer's suggestions and will revise the manuscript to better distinguish simulator-based robustness evaluation from hardware-based deployment validation, add the large-model evidence more clearly, and tighten the scope of our claims on CIM generalization.

---

> > ### Author Rebuttal · Reviewer_SnUP · 2026-04-03
> >
> > Thanks to the authors for the detailed rebuttal.
> >
> > Regarding the CIM simulator, my understanding is that it is a self-built, in-house simulator based on the manuscript description (“we built a CIM simulator, mimicking AWA (Büchel et al., 2022), and deployed the model on it for a year to test performance changes”). Based on this, I feel the claim that “the physical platform experiment primarily verifies deployment feasibility” is not fully accurate. In my view, the physical platform is also essential for validating the accuracy and fidelity of the simulator itself.

---

> > > ### Author Response · Authors · 2026-04-04
> > >
> > > We thank the reviewer for the careful follow-up and for pointing out that the statement “the physical platform experiment primarily verifies deployment feasibility” is not fully accurate. We agree with this observation and have accordingly refined our description of the relationship between the CIM simulator and the physical platform.
> > >
> > > Calibration between the simulator and the physical platform is important for assessing the reliability of simulation results, and that the physical platform can also provide supporting evidence for simulator fidelity. Ideally, access to more precise device-level noise and drift statistics would allow for a tighter alignment between the simulator and real hardware. However, in practice, the full distribution of hardware noise is difficult to obtain, especially in low-precision CIM systems, where noise arises from multiple sources including device mismatch, programming error, temporal drift, read noise, and peripheral circuitry. Therefore, we adopt a CIM-style noise/drift modeling approach widely used in prior work, which is intended to provide a reasonable and reproducible environment for long-term robustness evaluation, rather than claiming strict device-level calibration.
> > >
> > > To further support the validity and reasonableness of the simulator in our setting, we additionally conducted a direct alignment experiment, whose results are shown below.Specifically, we configured the CIM simulator to match the low-bit setting of the physical platform and evaluated the same deployed FashionMNIST model under the same noise level. Under this matched configuration, the simulator achieves an accuracy of 84.3%±1.11%, while the real physical platform achieves 82.93%±1.41%. The close agreement between these results provides additional supporting evidence that the simulator captures the main deployment-time behavior of the physical platform in this setting.
> > >
> > > | Setting            | Accuracy (%)    |
> > > |--------------------|-----------------|
> > > | CIM Simulator      | 84.3 ± 1.11     |
> > > | Physical Platform  | 82.93 ± 1.41    |
> > >
> > > Based on this, we will revise the manuscript to more accurately describe the role of the hardware experiments: they not only validate deployment feasibility but also provide empirical support for simulator fidelity. We will also include the above alignment experiment and corresponding discussion in the revised version to make the relationship between the simulator and the physical platform clearer, thereby improving the rigor and completeness of our work. We sincerely thank the reviewer for the helpful comments on the hardware deployment , which have helped us further strengthen this part of the paper.

---

### Decision · Program_Chairs · 2026-04-30

**Decision:**

Accept (regular)

**Comment:**

The paper received ratings of 4/4/4. The reviewers appreciated the technical contributions of the paper, while raising concerns about the theoretical proofs and scalability to large-scale experiments. The authors provided a rebuttal; although not all concerns were addressed, it improved the clarity. Overall, all reviewers leaned toward accepting the paper. The AC decided to accept it.